**Manuscript title**
Coffee and shade trees show complementary use of soil water in a traditional agroforestry
ecosystem
**Authors**
Lyssette E. Muñoz-Villers[1]*, Josie Geris[2], Susana Alvarado-Barrientos[3], Friso Holwerda[1], Todd E.
Dawson[4]
[1] Centro de Ciencias de la Atmósfera, Universidad Nacional Autónoma de México, Ciudad de
México, México
[2] Northern Rivers Institute, School of Geosciences, University of Aberdeen, Scotland, UK
[3] Red de Ecología Funcional, Instituto de Ecología, A.C., Xalapa, Veracruz, México
[4] Department of Integrative Biology, University of California-Berkeley, California, USA
*Correspondence to: Lyssette E. Muñoz-Villers, Centro de Ciencias de la Atmósfera, Universidad
Nacional Autónoma de México, Circuito Exterior s/n, Ciudad Universitaria, 04510 Ciudad de
México, México. Email: lyssette.munoz@atmosfera.unam.mx, Phone: (52) 55-5622-40-89.
*Abstract*
Globally, coffee has become one of the most sensitive commercial crops being affected by climate
change. Arabica coffee (*Coffea arabica*) grows in traditionally shaded agroforestry systems in
tropical regions and accounts for ~70% of the coffee production worldwide. Nevertheless, the
interaction between plant and soil water sources in these coffee plantations remains poorly
understood. To investigate the functional response of dominant shade trees species and coffee (*C.
arabica* var. *typica*) plants to different soil water availability conditions, we conducted a study during
a near normal and a more pronounced dry season (2014 and 2017, respectively) and a wet season
(2017) in a traditional coffee plantation in central Veracruz, Mexico. For the different periods, we
specifically investigated the variations in water sources and root water uptake via MixSIAR mixing
models that use $\delta^{18}O$ and $\delta^2H$ stable isotope composition of rainfall, plant xylem and soil water. To
further increase our mechanistic understanding about root activity, the distribution of belowground
biomass and soil macronutrients were also examined and considered in the model as prior
information. Results showed that, over the course of the two dry seasons investigated, all shade tree
species (*Lonchocarpus guatemalensis*, *Inga vera* and *Trema micrantha*) relied on average, on water
sources from intermediate (>15 to 30 cm depth: $58 \pm 18\%$ (SD)) and deep soil layers (> 30 to 120 cm
depth: $34 \pm 21\%$), while coffee plants used much shallower water sources (< 5 cm depth: $42 \pm 37\%$
and 5-15 cm depth: $52 \pm 35\%$). In addition, in these same periods, coffee water uptake was influenced
by antecedent precipitation, whereas trees showed little sensitiveness to antecedent wetness. Our
findings also showed that during the wet season coffee plants substantially increased the use of near
surface water (+56% from < 5 cm depth), while shade trees extended the water acquisition to much
shallower soil layers (+19% from < 15 cm depth) in comparison to drier periods. Despite the plasticity
in root water uptake observed between canopy trees and coffee plants, a complementary use of soil
water prevailed during the dry and wet seasons investigated. However, more variability in plant water
sources was observed among species in the rainy season when higher soil moisture conditions were
present and water stress was largely absent.

*Key words: Coffea arabica*; water stable isotopes, roots, nutrients, clay-rich soils, MixSIAR,
Mexico

## *1. Introduction*

Coffee agroforestry systems are highly valued because of their ecological, environmental, economic and social benefits (Mas and Dietsch, 2004; Perfecto et al., 2007; Tscharntke et al., 2011). Moreover, shade coffee of the species Arabica (*Coffea arabica*) accounts for ~ 70% of the total coffee production (USDA, 2017). Although Arabica coffee is mainly grown in tropical montane regions, it is cultivated under a wide range of climatic and soil conditions (Jha et al., 2014). Coffee Arabica plantations can be broadly classified as traditional or modern coffee systems, according to vegetation composition and structure and management practices (Moguel and Toledo, 1999). In the traditional systems, coffee plants are cultivated under a diverse canopy of native and/or introduced shade tree species. In contrast, monoculture coffee plantations exemplify the modern cultivation scheme, in which the shade is provided by a single commercial tree species. The use of agrochemicals is also typically required in this type of plantation (Moguel and Toledo, 1999).

Until recently, the vast majority of Arabica coffee was cultivated in traditionally managed shaded coffee plantations, which have lower production costs and enhanced biodiversity, carbon sequestration, soil fertility and biological pest control in comparison to modern systems (Greenberg et al., 1997; Perfecto et al., 2002; Kellermann et al., 2008). However, coffee management practices have become more intensive promoting the replacement of native trees with fast-growing monospecific timber species (i.e. *Cedrela odorata*, *Eucalyptus deplupta*, *Hevea brasilensis*) (Nath et al., 2011).

Growing a crop in association with shade trees inevitably leads to some degree of competition for the above-ground (light) and below-ground (water and nutrients) resources (Monteith et al., 1991). In an agroforestry system, the outcome of competition for light is relatively predictable due to the hierarchical structure of the canopy (i.e., shade trees intercept part of the sunlight, thereby reducing the amount available for the understory crop). Conversely, competitive interactions for below-ground resources can be much more diverse and complex. The central hypothesis of agroforestry underscores that crops and trees are complementary in their use of soil water (Cannell et al., 1996), however the degree to which this occurs will be largely controlled by the spatial and temporal patterns of resource availability, root distribution and root activity, which in turn depend on factors such as climate, soil conditions, crop and tree species, and plantation age, density and management practices (Beer et al., 1998; Lehmann, 2003; van Noordwijk et al., 2015). In addition, below-ground competitive interactions for water and/or nutrients are much more difficult to elucidate than above-ground relationships. So far, the most common approach is to measure the distribution of root abundance of crops and trees, and examine to what extent they overlap or are separated (e.g., Schaller et al., 2003; van Kanten et al., 2005). An important limitation of this method is, however, that the spatial

distribution of roots does not always mirror the actual resource capture along the soil profile (Dawson
et al., 2002; Lehmann, 2003). Another approach is to examine the vertical patterns of soil water
(Cannavo et al., 2011; Padovan et al., 2015) or nutrient (Schroth et al., 2000, cited in Lehmann, 2003)
depletion. However, these methods are problematic because they cannot provide information on
whether resource depletion is caused by the crop, the trees, or both (Cannavo et al., 2011; Padovan et
al., 2015). Recently, the use of hydrogen ($\delta^2$H) and oxygen ($\delta^{18}$O) water stable isotope techniques in
combination with mixing models based on Bayesian theory has proved to be a powerful tool for
quantifying the proportions and probability distributions of different water sources to plant uptake
across different ecosystems and regions (Barbeta et al., 2015; Beyer et al., 2018; Penna et al., 2018),
with the potential to largely overcome the above-mentioned limitations (Dawson et al., 2002;
Lehmann, 2003; van Noordwijk et al., 2015). Although rarely implemented, including nutrient and
root distribution data along the soil profile to inform these models could provide more comprehensive
insights into depth of plant water uptake (cf. Muñoz-Villers et al., 2018).
To date, research into plant-soil interactions and plant water source partitioning in coffee
agroforestry systems is extremely scarce. To our knowledge, only five studies have investigated the
water sources of shade trees and coffee shrubs using either information on the isotopic composition
of plant xylem and bulk soil water (Wu et al., 2016), soil water depletion (Cannavo et al., 2011;
Padovan et al., 2015) or root distribution (Schaller et al., 2003; van Kanten et al., 2005). Moreover,
all these studies have been carried out in intensive monospecific plantations characterized by high
coffee planting densities (~4000−5000 shrubs ha−1), low density (~150−280 trees ha−1) and very
low diversity (1-2 species) of shade trees. While recognizing the limitations of some of the methods
used in these previous studies, the available information suggests that competition for water between
coffee and trees can be strong at sites with a pronounced seasonal dry period (Wu et al., 2016;
Padovan et al., 2015), while it seems to be virtually absent at sites with no or a relatively short dry
season (Schaller et al., 2003; Cannavo et al., 2011). Further, although most coffee roots are usually
located in the upper soil layers (< 30 cm depth; van Kanten et al., 2005, and references therein), the
plant and soil interactions for water during the dry season seem to occur below the main crop rooting
zone (> 30 cm depth) (Wu et al., 2016). The latter reflects the ability of coffee to develop an extensive
root system, and to increase the root water uptake at greater soil depths once the available water has
been depleted in shallower layers (Huxley et al., 1974, cited in Lehmann, 2003).
Currently, we lack of information on plant water sources in traditional shade coffee
plantations. In these agroforestry systems, the higher density and diversity of shade trees could
potentially lead to stronger and more diverse tree-crop interactions (van Noordwijk et al., 2015). On
the other hand, the dense tree canopy reduces light availability and hence limits coffee water use. This
could lead to a lower soil water demand and thus increased plant water availability during the dry
season.

Further, ecohydrological research in these shade coffee systems is becoming increasingly
important since trees have been promoted as a strategy for mitigating and adapting to future climate
(Schroth et al., 2009; Vaast et al., 2016; Rice, 2018). Shaded coffee plantations store more carbon
than sun-grown coffee systems, thereby contributing to the reduction of greenhouse gases (Vaast et
al., 2016; Rice, 2018, and references therein). In addition, the tree canopy provides some level of
protection against the rising mean and maximum air temperatures (Baker and Haggar, 2007; Schroth
et al., 2009; Vaast et al., 2016), which in recent modeling studies have been pointed out as the key
climatic changes affecting coffee growth, yield and quality (Schroth et al., 2009; Baca et al., 2014;
Bunn et al., 2015). Although there are important differences across sites, rainfall is also predicted to
decrease and become more variable in many of the world's coffee-growing regions. For example,
Giorgi (2006) estimated that rainfall will decrease by about 17% (per 100 years) during the dry season
and by about 9% during the wet season in Mexico and Central America. Similarly, predictions by
Karmalkar et al. (2011) for the same regions pointed out changes in rainfall of −24% to +8% (per 100
years) during the dry season and of −39% to −1% during the wet season. As such, if warming is
accompanied by decreases in rainfall, this could lead to, or exacerbate, competition for water sources
between coffee shrubs and shade trees (Baker and Haggar, 2007), which in turn could affect the long-
term sustainability of these agroecosystems.

Mexico is among the largest shade coffee producers in the world, and the central region of
Veracruz constitutes the second most important coffee zone in the country. In this area, we selected
a representative traditional shade coffee plantation to investigate plant water sources of dominant
shade trees species and coffee (*C. arabica* var. *typica*) shrubs under different conditions of soil water
availability. During a near normal and a more pronounced dry season (2014 and 2017, respectively)
and a wet season (2017), variations in depth of plant water uptake were examined using the stable
isotopic composition ($\delta^{18}$O and $\delta^2$H) of rainfall, plant xylem and soil water in combination with a
Bayesian mixing model (MixSIAR), along with microclimatic and soil moisture measurements. To
further increase our understanding about root activity and water uptake, the distribution of roots and
macronutrients along the soil profile were also examined and considered in the mixing model as prior
information. Specifically, we addressed the following questions:

1.  Does a complementary water use strategy between shade trees and coffee shrubs prevail
over competition in a traditional shaded agroforestry system?

2. Does competition exist for water sources among tree and coffee species during more

pronounced dry periods?

3. What are the seasonal patterns in plant-water source partitioning?

***2. Materials and methods***
*2.1 Study site*

The research was carried out in the "La Orduña" coffee plantation (~100 ha) located on a flat

plateau at an elevation of 1210 m a.s.l. on the eastern slopes of the Cofre de Perote mountain (19°28′
N, 96°56′ W) in central Veracruz State, Mexico (Fig. 1). The coffee plantations in this region occur
between elevations of 1000 and 1350 m a.s.l. (Hernández-Martínez et al., 2013; Marchal and Palma,

1985).

The climate is classified as temperate humid with abundant rains during the summer (García,

1988). Two distinct seasons can be distinguished: (1) a wet season (May–October), during which
rainfall is associated primarily with cumulus and cumulonimbus clouds formed during convective
and orographic uplift of the moist maritime air masses brought in by the easterly trade winds; and (2)
a (relatively) dry season (November–April), during which most rainfall falls from stratus clouds
associated with the passage of cold fronts (Báez et al., 1997). Mean annual rainfall measured nearby
the study site during the period 1971−2000 was 1765 mm, with on average 389 mm falling during
the dry season and 1376 mm falling during the wet season (SMN, 2018). Mean annual temperature
over this period was 19.5 °C, with a minimum and maximum monthly average value of 15.5 and
22.5°C observed in January and May, respectively (SMN, 2018). Annual potential evapotranspiration
($ET_0$) is about 1120 mm (Holwerda et al., 2013).

The investigated shade coffee plantation is a so-called traditional commercial polyculture

system (*sensu* Moguel and Toledo, 1999), which was established more than 80 years ago. The tree
canopy was diverse and consisted predominantly of the species *Inga spp.*, *Citrus spp.*, *Lonchocarpus*
*guatemalensis*, *Trema micrantha* and *Enterolobium cyclocarpum* (Holwerda et al., 2016). The shade
trees were planted at a density of ca. 500 ha−1, and currently form a canopy of about 14 m high. The
Arabica coffee plants were of the variety *typica*. *Typica* −a tall cultivar of *Coffea arabica*− was the
first coffee variety that arrived from Ethiopia to Mexico (Renard, 2010); it has bronze-tipped young
leaves and the berries are large. Plants of *typica* variety are tolerant to conditions of low soil fertility
and drought, but vulnerable to most pests and diseases (Escamilla et al., 2005). In the study site, this
cultivar was planted approximately 20 years ago at a density of about 1700 shrubs ha−1, currently
having an average height of ~ 2 m. In this region, the coffee flowering occurs in March or April, fruit
development between May and October, and ripening and harvest between October and February
(Villers et al., 2009). The management of the plantation involves weed control practices and selective
pruning of mature coffee plants and shade trees at irregular times once every ~ 7 years (cf. Hernández-
Martínez et al., 2009). No pruning activities occurred during or in between our study periods. A
photograph of the coffee plantation is provided in the Supplementary Material.

The soil type is an Andic Acrisol derived from volcanic ashes. Soil profiles (~150 cm) are

multilayered (A, B1/BT and BC) and have clay (~ 65%) as the dominant texture across all layers. A
general description of the soil profile showed a dark brown to dark yellowish brown, clay silty organic
A horizon (0–20 cm) overlying a dark yellowish brown, clay silty sand B1/BT horizon (20–135 cm),
followed by a dark yellowish brown, clay sandy BC horizon (>135 cm). Average soil bulk densities
and porosities were 1.2 gr cm−3 and 63%, respectively, along the A and B horizons (Holwerda et al.,
2013). The underlying material consists of deeply weathered old lava and sandy-gravelly pyroclastic
flow deposits (Rodríguez et al., 2010). Soils were mostly covered by a thin (1-2 cm) but continuous
layer of litter.

*2.2 Hydrometeorological measurements*

During the study period, rainfall and microclimate conditions were continuously monitored

above the canopy in an 18 m high tower, located in the southwestern part of the coffee plantation.
Rainfall ($P$, mm) was measured using a TR–525 M tipping bucket rain gauge (Texas Electronics,
USA). Temperature ($T$, °C) and relative humidity (RH, %) were measured using a HC2-S3 probe
(Rotronic, USA). Data were recorded every 30 s, accumulated ($P$) or averaged values (all other
parameters) were stored at 5-min intervals using a CR1000 datalogger (Campbell Scientific Ltd.,
USA).

*2.3 Isotope sampling*

To examine the water sources of overstory shade trees and understory coffee shrubs, plant

tissue and soil samples were collected for isotope analysis at the middle (Jan. 23) and end (Apr. 11
and 26) of the 2014 dry season. In 2017, the dry season was warmer and drier offering the opportunity
to examine the vegetation responses to more pronounced dry conditions. Therefore, a second
sampling campaign was carried out to collect plant and bulk soil samples at the middle (Feb. 27), end
(Apr. 5) and late end (May. 20) of the 2017 dry season. Another sampling was carried out in the
middle of the 2017 wet season (Aug. 4) to evaluate plant-soil water uptake patterns at higher soil
water availability conditions.

In all seven samplings, xylem samples were obtained from three individuals of each of the

three dominant shade tree species (*Lonchocarpus guatemalensis*, *Inga vera* and *Trema micrantha*) by
extracting ~5-6 cm cores using a Pressler increment borer inserted at 1.2 m above ground ($n = 60$
samples of trees in total). On each occasion, xylem samples were taken from the same individuals but
from various aspects of the trunk. The bark was immediately removed after core extraction to avoid
contamination of phloem water. For the coffee plants, samples were obtained from ~6 cm segments
of mature suberized branches that were cut near the main stem of several shrubs each time. The bark
(~1mm thick) and cambium were not stripped from the coffee branches, to avoid exposure of the
samples to evaporation. All coffee plants were sampled randomly ($n = 40$ samples of coffee shrubs
in total). During the 2014 and 2017 dry seasons, sampling of coffee shrubs involved 5-6 individuals
each time. Since only one sampling occasion was performed during the 2017 wet season, a larger
number of individuals (10) was sampled to reduce the uncertainties associated with different sampling
sizes between wet and dry seasons respectively. For each tree, we measured diameter at breast height
(DBH) and height, and for the coffee plants the diameter of the main stem was measured below its
bifurcation in small branches (Table 1).
Bulk soil samples were collected at three locations and at depths of 5, 15, 30, 60, 90 and 120
cm, using a hand auger ($n = 126$ samples of soil in total). Auger sampling points were located so that
each of the sampled shade trees and coffee plants had one soil sampling point within a 3 m radius.
Samples of xylem and bulk soil were collected during the morning and early afternoon
(between 8:30 to 13:30 hrs), and each sampling campaign was preceded by at least 6 days up to 22
days without or with minimum accumulated rainfall (< 5 mm). All xylem and soil samples were
collected quickly and carefully and stored in water-tight vials to avoid any evaporation (see section
below).
To establish the local meteoric water line and compare soil water sources with recent rainfall,
bulk samples of rainfall ($n = 80$ in total) were collected weekly at a nearby (~ 5 km) meteorological
station over the course of the two years studied (Nov. 2013 – Oct. 2014 and Nov. 2016 – Oct. 2017)
as part of a long-term isotope sampling of precipitation (cf. Muñoz-Villers et al., 2018).

*2.4 Isotope collection and analysis*
Samples of precipitation, plant xylem and bulk soil for isotope analysis were collected in 30-
ml borosilicate glass vials sealed with polycone caps to prevent evaporation. All samples were
refrigerated until extraction and analysis at the Center of Stable Isotope Biogeochemistry (CSIB) at
the University of California-Berkeley, USA.
Xylem and soil samples were extracted using cryogenic vacuum distillation (temperature:
$100 \pm 1.1°C$, vacuum: $3 \pm 1.5$ Pa and time: 60-70 min) following the method of West et al. (2006).
The $\delta^2H$ and $\delta^{18}O$ isotopic compositions of extracted water samples were then determined using an
isotope-ratio mass spectrometer (Thermo Delta Plus XL, Thermo Fisher Scientific, USA). The
analytical precision of the instrument was $\pm$ 0.60‰ (1 SD) for $\delta^2$H and $\pm$ 0.12‰ (1 SD) for $\delta^{18}$O.
Samples of precipitation were analyzed for $\delta^2$H and $\delta^{18}$O using a laser water isotope analyzer (L2140-
i) from Picarro Inc. (Santa Clara, CA, USA) in high precision and without Micro-Combustion Module
mode. The analytical precision was $\pm$ 0.65‰ (1 SD) and $\pm$ 0.20‰ (1 SD) for $\delta^2$H and $\delta^{18}$O,
respectively.

The isotope values are expressed in delta notation (‰) relative to Vienna Standard Mean

Ocean Water (VSMOW). To evaluate evaporative enrichment in the soil and xylem water isotopes
relative to rainfall, we calculated the deuterium-excess parameter ($d = \delta^2$H - 8 * $\delta^{18}$O; Dansgaard,

1964).


*2.5 Soil sampling and laboratory determinations*

To determine volumetric soil water content (SWC), samples were collected at 5, 15, 30, 60,

90 and 120 cm depth from each of the three boreholes excavated during the soil isotope samplings.
Soil moisture content was determined gravimetrically and converted to volumetric values by using
bulk density of the soil sample. In addition, to determine the antecedent moisture conditions for the
15 days prior to each sampling date, an antecedent precipitation index (API) was calculated following
Viessman et al. (1989).

To examine pH and N, P and K macronutrient concentrations along the soil profile, soil

samples were collected at 5, 15, 30, 60, 90 and 120 cm depth from each borehole ($n$ = 3 samples per
soil depth) during three isotope sampling campaigns: Apr. 11, 2014 (dry season), Feb. 27, 2017 (dry
season) and Aug. 4, 2017 (wet season). Samples ($n$ = 18) for determining other chemical properties
were collected at the same depths in soil profiles. All samples were first air-dried and then sieved
using 2 mm screens. Soil pH was determined using a glass electrode pH meter in a 1:2 soil: water
ratio. Organic matter (OM) was determined by the Walkley-Black method. Total carbon (C) and total
nitrogen (N) were measured using a TruSpec dry combustion CN analyzer (LECO, USA). Extractable
phosphorus (P) was determined by the Bray I method (Bray and Kurtz, 1945). Exchangeable cations
(Ca+, Mg+, K+, Na+) were determined by extracting soil with 1 MNH4OAc (pH 7.0). Ca+ and Mg+
were analyzed using atomic absorption spectrometry and K+ and Na+ were analyzed using flame
photometry. Soil cation exchange capacity (CEC) was determined by the ammonium acetate 1N (pH
7.0) method (Van Reeuwijk, 2002) and base saturation (BS) was calculated as the portion of CEC
that is occupied by exchangeable bases: (Ca+, Mg+, K+, Na+)/CEC.

*2.6 Root biomass*

To examine the root biomass distribution along the soil profile in the study plot, 33 soil cores

were collected using 5 cm diameter and 10 cm long samplers. Soil cores were extracted at 5, 20, 40,

60 and 90 cm depth (from 5 to 40 cm: $n = 9$ for each depth, and from 60 to 90 cm: $n = 3$ for each

depth). All cores were processed immediately in the laboratory. Soil samples were first sieved using

2 mm screens to separate the bigger roots. Next, the samples were washed using a fine nylon mesh

sieve, and then separated into diameter classes (< 1 mm, 1−2 mm and > 2 mm) and dried at 70 °C for

48 hours. Root biomass (g m−3) was calculated from the dry weight of the roots and the volume of

the core sampler for each class and soil depth. No differentiation between roots of coffee shrubs and

shade trees was made.

***2.7 Plant water uptake sources and temporal patterns***

The MixSIAR Bayesian mixing model framework (Moore and Semmens, 2008; Stock et al.,

2018) was used to determine the most likely contributions of water sources for the shade tree species

and coffee shrubs sampled over the course of the 2014 (Jan. 23, Apr. 11 and 26) and 2017 (Feb. 27,

Apr. 5, May. 20) dry seasons and the 2017 wet season (Aug. 4). To assess temporal changes of the

different plant water sources, the seven sampling occasions were modeled separately. The mixture

data for the model was the mean xylem water isotopic ($\delta^2$H and $\delta^{18}$O) composition of the shade tree

species and coffee shrubs, changing accordingly with the sampling date. Based on statistical tests, the

relative contributions of four potential plant water sources were evaluated and restricted to the

following soil groups: near surface water (< 5 cm), shallow (5 to 15 cm), intermediate (> 15 to 30 cm)

and deep soil water (> 30 to 120 cm). For each sampling date, the mean and standard deviation of the

soil water isotope ($\delta^2$H and $\delta^{18}$O) signatures from the four different grouped soil depths were

introduced into the model, all corresponding to the date of xylem tissue collection.

Further, we also considered the use of additional data such as soil macronutrients (N, P, K)

and root biomass information to constrain model estimates by specifying an 'informative' prior

distribution of the soil source proportions (Stock et al., 2018). These data were also grouped into four

classes based on the depth of the soil samplings and corresponding largely with the grouping for soil

water: near surface (< 5 cm) shallow (5 to 20 cm), intermediate (> 20 to 40 cm) and deep (> 40 to

120 cm). In addition, the nearest corresponding dry or wet season dataset of soil macronutrients were

used according to the date of sampling. More details on the informative prior parametrization are

provided in the Supplementary Materials. The effect of using these priors (i.e. a weight proportion

before considering the isotope data) on the water sources distribution was then examined by

comparing these with the results of 'non-informative' (i.e. all the combinations of proportions of

water sources were equally likely) simulations. The results of each of these model runs were accepted

based on the examination of Markov Chain Monte Carlo convergence using the Gelman-Rubin and
Geweke diagnostic tests (Gelman et al., 2014).
Furthermore, the effect of isotope fractionation on the quantification of plant water sources
was specifically explored by comparing the results of the informed two-isotope mixing model with
those from a mixing model using only one water stable isotope ratio in the MixSIAR Bayesian
framework. This approach has been used elsewhere (e.g. Evaristo et al., 2017; Barbeta et al., 2019)
to provide some initial insights. Nevertheless, we are aware that the use of a single isotope ratio
approach in a multiple water source model could lead to erroneous results due to the overlap of
feasible solutions with poor constrained of uncertainties (see Parnell et al., 2010).
Lastly, the relative contributions of the water sources were compared among shade trees and
coffee shrubs across all sampling dates using factorial ANOVA and Tukey's HSD post-hoc tests. The
analyses were carried out in R Statistical Software version 3.2.4 (R Core Development Team, 2016).

### 3. Results

### 3.1 Hydrometeorological conditions

Precipitation ($P$) was 1650 mm in the first study year (Nov. 2013 – Oct. 2014) and 1423 mm
in the second study year (Nov. 2016 – Oct. 2017). During the 2013-2014 dry season (Nov – Apr.),
rainfall was 323 mm, and mean daily values of temperature ($T$) and vapor pressure deficit (VPD)
were $17.6 \pm 3.0°C$ and $0.65 \pm 0.39$ kPa, respectively. The lowest monthly $P$ and the highest $T$ and
VPD were observed in April at the end of the dry season (Fig. 2a,b). During the 2016-2017 dry season,
rainfall amounted to 235 mm, with lowest monthly values registered in January and February at the
middle of the season (Fig. 2b). Mean daily $T$ was $18.3 \pm 2.6°C$, with the highest values observed at
the end of the dry period. Generally, VPD was high during the entire dry season ($0.78 \pm 0.46$ kPa on
average), and reached maximum values in February and May.
Compared to long-term (1971−2000) climatic records of the region, rainfall in the first study
year was very close to the mean annual precipitation of 1765 mm (SMN, 2018). In contrast, the second
year was drier (~ 300 mm less; –20%), especially during the dry season, which had about 40% lower
precipitation than the average value of 389 mm. Also, higher mean monthly temperatures (+ 0.54°C)
prevailed across the 2017 dry season in comparison with the 1971−2000 period. Although rainfall
during the 2013-2014 dry season was also about 20% lower than normal, this season was considered
as near average.
Rainfall during the 2017 wet season (May – Oct.) was lower in comparison to 2014 (1188
mm vs. 1326 mm, respectively) (Fig. 2b). Further, the mean air temperature and vapor pressure deficit
were slightly higher in the 2017 wet season than in the 2014 wet season (20.7 ± 1.6°C and 0.67 ±
0.25 kPa vs. 20.1 ± 1.5°C and 0.60 ± 0.21 kPa, respectively) (Fig. 2a).

### 3.2 Soil moisture and antecedent precipitation during sampling campaigns

During the 2014 dry season campaign (Jan. – Apr.), mean soil water content (SWC) was on
average 33.8 ± 1.7% at 5 cm depth, 40.2 ± 14.5% at 15 cm depth, 38.9 ± 6.4% at 30 cm depth and
48.3 ± 1.4% at 60 to 120 cm depth (Fig. 2b). In comparison, SWC in the 2017 dry season campaign
(Feb. – May.) was lower in the first 30 cm (32.5 ± 3.9%), meanwhile water content in the deeper
layers was similar (49.0 ± 2.9%) with respect to the 2014 dry period. In 2014, lowest SWC values
were observed at the end of the dry season (April), whereas the greatest soil moisture depletion in
2017 was registered at the middle of the dry season (February) (Fig. 2b).
During the wet season sampling in August 2017, SWC values at 5 cm (28.2 ± 2.6%), 15 cm
(30.9 ± 4.3%), 30 cm (38.4 ± 4.8%) and 60 to 120 cm (49.0 ± 2.9%) depths were generally higher in
comparison to the 2017 dry period (Fig. 2b). Although the 2017 wet season sampling showed slightly
lower SWC values in the shallower soil layers in comparison to the 2014 dry season, the SWC values
in the deeper layers were higher. For the different samplings, antecedent precipitation conditions
(API) were, respectively, 4, 30 and 13 mm for Jan. 23, Apr. 11 and 26, 2014 and 1, 12, 9 and 43 mm
for Feb. 27, Apr. 5, May. 20 and Aug. 4, 2017.

### 3.3 Stable isotope composition of waters

Over the study periods, a greater range of variation was found in the rainfall isotope
composition of the 2013-2014-year (from –126.7 to 14.4‰ for $\delta^2H$; from –17.7 to 0.0‰ for $\delta^{18}O$) in
comparison to the 2016-2017-year (from –113.3 to 15.5‰ for $\delta^2H$; from –15.9 to 0.0‰ for $\delta^{18}O$) ($p$
> 0.05) (Fig. 3). Overall, mean dry season rainfall was significantly more enriched than the mean wet
season rainfall in $\delta^2H$ and $\delta^{18}O$ ($p \leq 0.001$) (Table 2 and 3). On average, the isotopic compositions of
the dry and wet season rainfall were both more depleted during the second study year than during the
first study year; thus, the local meteoric water line of 2016-2017 had a slightly steeper slope in
comparison to the one for 2013-2014 (Fig. 3). Nevertheless, the range of variation of deuterium excess
values was similar between years (9–29‰ for the first year vs. 9–31‰ for the second year; Fig. 3),
and deuterium excess values of rainfall within the dry and wet seasons were not statistically different
($p \geq 0.05$).
For all sampling dates, hydrogen and oxygen isotope composition of bulk soil water showed
a consistent pattern of increasing isotope depletion with soil depth (Supplementary Materials), in
which shallower (5-15 cm) soil water was significantly more enriched than intermediate (15-30 cm)
and deeper (30-120 cm) soil water layers ($p \leq 0.001$) (Table 2 and 3; Fig. 3). In correspondence,
lowest values of deuterium excess generally characterized the near surface soil water pool.

For the 2014 dry season samplings, bulk soil ranged from –83.3 to –11.9‰ for $\delta^2$H and from

–11.1 to –0.9‰ for $\delta^{18}$O (Fig. 3a). For the 2017 dry season samplings, bulk soil water showed a
narrower range of variation and more enriched isotope values (from –54.8 to –19.1‰ $\delta^2$H and from
–7.5 to –1.5‰ $\delta^{18}$O) in comparison to 2014 (Fig. 3b). However, statistical differences were only
suggested for the intermediate and deeper soil layers in both water isotopes between the two dry
seasons investigated ($p \leq 0.001$).

In the 2017 wet season sampling, bulk soil isotope composition ranged from –70.5 to –37.5‰

for $\delta^2$H and from –8.4 to –4.1‰ for $\delta^{18}$O (Fig. 3c), showing significant differences in the shallow,
intermediate and deep soil water pools in comparison to the 2017 dry season ($p \leq 0.001$). In all
sampling periods, bulk soil water across the different depth groups was isotopically distinct from
rainfall during the 2014 and the 2017 dry seasons ($p \leq 0.001$ for both water isotopes).

Across all sampling periods, xylem water of coffee shrubs was more enriched than that of

shade trees ($p \leq 0.001$) (Table 2 and 3; Figure 3). In the 2014 dry season, xylem water isotope values
of shade trees ranged from –65.5 to –32.1‰ for $\delta^2$H and from –7.6 to –3.6‰ for $\delta^{18}$O, meanwhile a
larger variation was observed in the xylem water of coffee shrubs (from –46.5 to –9.6 ‰ $\delta^2$H and
from –6.3 to –0.6‰ $\delta^{18}$O) ($p \leq 0.001$) (Fig. 3a). Among tree species, *Lonchocarpus guatemalensis*
showed the most depleted xylem water isotope signature (–58.1 ± 4.8‰ $\delta^2$H and –6.8 ± 0.5‰ $\delta^{18}$O),
whereas *Inga vera* had the most enriched values with a greater range of variation (–51.0 ± 10.2‰ $\delta^2$H
and –5.3 ± 1.1‰ $\delta^{18}$O). Statistical tests showed that *Inga vera* was different from the other tree species
in $\delta^{18}$O ($p < 0.05$).

In the 2017 dry season, the isotopic composition of shade trees varied from –56.7 to –34.5‰

for $\delta^2$H and from –6.0 to –3.2‰ for $\delta^{18}$O; corresponding values for coffee shrubs varied from –39.6
to –7.8 ‰ for $\delta^2$H and from –4.4 to –1.1‰ for $\delta^{18}$O ($p \leq 0.001$) (Fig. 3b). Contrary to 2014, *L.*
*guatemalensis* showed the most enriched isotope value (–41.3 ± 5.7‰ for $\delta^2$H and –4.6 ± 0.5‰ for
$\delta^{18}$O), and *I. vera* had the most depleted values (–48.5 ± 5.1‰ for $\delta^2$H and –4.8 ± 0.8‰ for $\delta^{18}$O),
with differences being statistically significant for $\delta^2$H ($p < 0.05$).

Overall, isotope values of plant xylem water were more enriched during the 2017 dry season

than during the 2014 dry season ($p \leq 0.001$) (Fig. 3a,b; Fig. 4). Deuterium excess values were also
lower in shade trees and coffee shrubs during 2017, indicating a more evaporative signature (Table 2
and 3; Fig. 3). Plots of $\delta^2$H xylem water against height for the individual shade trees and coffee shrubs
sampled in both dry seasons are shown in Figure 4, in which a similar $\delta^2$H pattern was displayed
between trees and coffee shrubs in the 2014 and 2017 years.

During the 2017 wet season sampling, $\delta^2H$ and $\delta^{18}O$ values in xylem water of trees and coffee

shrubs were more depleted in comparison to the 2017 dry season ($p < 0.05$) (Fig. 3c). The range of
variation was from –60.6 to –45.6 ‰ $\delta^2H$ and –6.2 to –5.4‰ $\delta^{18}O$ for trees, and from –42.2 to –34.4
‰ $\delta^2H$ and –5.4 to –4.4‰ $\delta^{18}O$ for coffee shrubs ($p \leq 0.001$).

It was observed that the xylem isotopic composition of all shade trees and coffee plants fell

within the range of the soil water sources during the 2014 dry season samplings (Fig. 3a). For the
2017 dry season, we again observed a good isotopic match between the shade tree xylem water and
soil water. However, for the coffee plants, the xylem water was more enriched in $\delta^2H$ in comparison
to soil water (Fig. 3b). During the 2017 wet season sampling, a slight enrichment in $\delta^2H$ was again
observed in the xylem water of coffee, while trees showed a good overlap with soil water (Fig. 3c).
Based on these results, tests were carried out to specifically evaluate the effects of deuterium
fractionation on coffee water sources by running a simple mixing model using only hydrogen isotope
ratios in the MixSIAR framework.

### 435    *3.4 Root biomass and macronutrients along soils profile*

Overall, most roots were concentrated in the first 5 cm of soil with a sharp decline in biomass

at 20 cm depth (Fig. 5a). Fine roots (< 1mm) followed by bigger roots (> 2 mm) dominated the
shallower soil layers (< 20 cm), meanwhile roots in general were scarce at deeper depths (> 60 cm).
Soil acidity was highest near the surface and decreased gradually with depth (Table 4). Organic matter
(OM) and total carbon were also greatest between 5 and 15 cm depth, while values decreased rapidly
below ~30 to 60 cm depth. Although highest concentrations of nitrogen were found in the first 15 cm
of soil, values remained relatively high and constant at deeper layers (Fig. 5b). Phosphorus showed
its highest concentration at the topsoil with values decreasing sharply below 30 cm depth. In contrast,
concentrations of potassium, sodium and magnesium were lowest in the first 15 cm, while maximum
values were observed below 90 cm depth. Base saturation (BS) was very low along the soil profile,
indicating poor availability of soil macronutrients. Soil cation exchange capacity (CEC) was generally
low across depths, indicating little potential for interaction between clay particles and cations.

### 449    *3.5 Plant water sources*

We found a good agreement between the MixSIAR Bayesian mixing model results using a

non-informative and an informative prior distribution (on average 5% difference across all xylem
water contributing sources; $p > 0.05$). This indicates that the independent distribution (soil
macronutrients and root data) set a *priori* to optimize source proportion estimates (informative
approach) in the model was not influential enough to significantly modify the results obtained using
the isotope signatures of the xylem water sources alone (non-informative approach). Having this
agreement between models, we present the results of the water source contribution based on the
informative model runs. Results of the non-informative approach are provided in the Supplementary
Materials.

The model results showed that the intermediate and deep soil water pools ($> 15$ to 120 cm
soil depth) were the main sources for the shade trees over the course of the 2014 dry season ($91 \pm$
37% on average; Fig. 6 and Supplementary Materials). Across this period, *L. guatemalensis* showed
on average the highest proportion of water uptake between 30 and 120 cm soil depth ($49 \pm 26\%$),
while *T. micrantha* and *I. vera* depended strongly on soil water sources between 15 and 30 cm ($54 \pm$
18% and $67 \pm 6\%$) ($p < 0.001$). In contrast, the water uptake of coffee plants was mainly sustained by
sources from the first 15 cm of soil ($94 \pm 27\%$ on average; Fig. 6 and Supplementary Materials),
having significant differences with all shade tree species ($p < 0.001$).

During the 2017 dry season, the same trend with most water extracted from intermediate and
deep soil layers was observed in the shade trees ($91 \pm 39\%$ on average; Fig. 7a,b,c and Supplementary
Materials). Among samplings dates, differences between tree species only appeared to occur at the
end of the dry period (Apr. 5) ($p < 0.05$). Coffee water sources were again restricted to much shallower
soil layers (0–5cm: $53 \pm 44\%$ and 5–15 cm: $42 \pm 41\%$; Fig. 7a,b,c and Supplementary Materials)
compared to shade trees.

Overall, we did not find any statistically significant difference among main plant water
sources between the dry periods investigated ($p > 0.05$). Across the individual samplings throughout
the two dry seasons, we observed that antecedent precipitation had a stronger effect on the water
uptake sources of coffee plants than trees (Fig. 8). For example, when dry antecedent wetness
prevailed ($API_{15} < 5$ mm; Fig. 2b) coffee water sources were mainly composed of soil water from $>$
5 to 15 cm depth ($91 \pm 3\%$). Alternatively, when wetter antecedent conditions were present ($API_{15} >$
10 mm), the near surface soil water layer ($58 \pm 31\%$) was the main contributing source. On the
contrary, tree water uptake was essentially sustained by deeper soil water sources at low and relatively
high antecedent wetness conditions ($94 \pm 23\%$ and $87 \pm 23\%$, respectively) (Fig. 8). Nevertheless, for
all species investigated, the relationships between API and the contribution of near surface soil water
sources were not statistically significant ($p > 0.05$).

During the 2017 wet season, water source partitioning differed among shade tree species
(Fig. 7d and Supplementary Materials). During this period, *L. guatemalensis* and *I. vera* showed the
greatest use of deep soil water ($74 \pm 37\%$ and $69 \pm 41\%$, respectively), while shallower soil water
was the main source for *T. micrantha* ($91 \pm 23\%$), having significant differences with the other tree
species ($p < 0.001$). Coffee consistently showed the use of near surface water sources ($98 \pm 5\%$; Fig.
7d and Supplementary Materials), which was significantly different from all shade tree species ($p < $

0.001).


### 3.6 Fractionation effects on coffee water sources

To evaluate the effects of xylem deuterium fractionation on our results for coffee water source
uptake, we compared the relative contribution of each soil water source obtained via the single-
isotope ($\delta^2$H) mixing model with those obtained via the informative two-isotope mixing model. In
general, we observed that the $\delta^2$H model consistently estimated a lower contribution of the shallow
soil water source and a higher contribution of the near surface soil water source (Supplementary
Materials). On average, the reduction in the shallow soil water source ($-25.7 \pm 29.0\%$) coincided very
well with the increase in the near surface soil water source ($+28.1 \pm 30.6\%$). These differences were
most pronounced for the 2017 dry season samplings ($p > 0.05$; Supplementary Materials), during
which the differences in $\delta^2$H between coffee xylem water and soil water were greatest. However,
there were no significant differences between the relative contributions of the intermediate and deep
soil water sources estimated by the two models ($p > 0.05$). In summary, the results of the $\delta^2$H mixing
model suggested an even more pronounced soil water partitioning between coffee and shade tree
species than those obtained with the informative two-isotope mixing model.

### 4. Discussion

### 4.1 Methodological aspects

To our knowledge, the ecohydrological study presented here is one of the first that
incorporates biophysical properties as prior information alongside plant water source information
from stable isotope ($\delta^{18}$O and $\delta^2$H) data into a MixSIAR Bayesian mixing model framework, as a way
to improve our understanding of the processes that lead to differences in the depth of plant water
uptake. Even though our findings did not change significantly by including or excluding the prior
information such as soil macronutrients and root data, exploring plant water source partitioning using
these two model approaches provided more confidence in our results. Therefore, we call for more
studies that combine soil nutrient and root biomass distribution with plant water source information
from $\delta^{18}$O and $\delta^2$H data, to explore the additional value of these biophysical parameters elucidating
plant-soil interactions in different regions and environments.
In recent years, some plant, soil and/or deep subsurface water source studies that have used
stable isotopes have identified isotope variation that could be the result of isotope fractionation
processes caused by water molecules interacting with clay surfaces, partially filled pore spaces or
even salts (Chen et al., 2016; Gaj and McDonnell, 2019; Lin et; al., 2017Oerter et al., 2014; Oshun
et al., 2015). Our soils were rich in clay content and according to some studies this type of soil
structure can impart isotope fractionation (Lin et al., 2017; Meißner et al., 2014; Oerter et al., 2014;
Orlowski et al., 2016a). Thus far, however, these isotope effects have been more evident in clay-rich
soils having high cation exchange capacities (CEC ~ 30 to 70 $cmol_c$ kg–1; Oerter et al., 2014;
Orlowski et al., 2016b) in combination with low soil water contents (SWC < 20% Meißner et al.,
2014; Orlowski et al., 2016b). In this respect, the soils in our study area are characterized by low CEC
(< 21 $cmol_c$ kg–1; Table 4). This reflects relatively little interaction between cations adsorbed and
clay mineral particles, which indirectly suggests minimal impacts of interlayer water bound in the
soil structure (cf. Vidal and Dubacq, 2009). In addition, our soil samples were collected at relatively
high SWC across the different sampling periods (~ 30% to 60%; Figure 1). As such, we have assumed
that the probability of fractionation due to soil properties that may impact water extraction efficiency,
was very small or completely absent and therefore, the extracted soil water was the same the plants
had access to.
With regard to our plant samples, we specifically observed enrichment in the deuterium
composition of the xylem water in the coffee plants in comparison to bulk soil water. It is not
surprising that fractionation was evident for $\delta^2H$ and not $\delta^{18}O$, given the higher fractionation factor
of $^2H$ relative to $^{18}O$ (Rundel et al., 2012). Some possible explanations for this xylem water
enrichment could be related to bark evaporation (Ellsworth and Sternberg, 2015) and/or xylem-
phloem water exchange (Cernusak et al., 2005), since we did not remove the bark and cambium from
our coffee branch samples. On the other hand, like many other crops, coffee plants associate
symbiotically with arbuscular mycorrhizal fungi (López-Andrade et al., 2009; Perea-Rojas et al.,
2019). Studies in our coffee growing region of Veracruz have documented the presence of
mycorrhizal structures in coffee roots (Arias et al., 2012; Muleta et al., 2008), which can promote
increases in plant water and nutrient uptake (Augé, 2004; Scheneiger and Jakobsen, 2000). Although
no research has been carried out yet to test the influence of mycorrhizal fungi on isotope fractionation
during coffee root water uptake, this effect could have been present and being also responsible for the
isotopic mismatch between the coffee xylem water and soil water sources, as it has been reported
elsewhere (Poca et al., 2019).
We did evaluate the effects of these isotope enrichments in the coffee xylem water on the
relative contributions of the coffee water sources using a single-isotope ($\delta^2H$) mixing model.
Consistently, the model results estimated a higher near surface water and a lower shallow soil water
source contribution in comparison to the dual isotope informative prior mixing model. In contrast,
the estimated proportions of the intermediate and deep soil water sources were similar between
models. Thus, the effect of fractionation was translated into a more pronounced spatial separation
between the main soil water sources of the coffee plants and shade trees, but our overall results were
not different.

### 4.2 Complementary water use strategy between shade trees and coffee shrubs

Our findings showed that all shade tree species (*L. guatemalensis*, *I. vera* and *T. micrantha*)

relied mainly on water sources from deep soil layers (> 15 to 120 cm depth), while the use of much
shallower water sources (< 15 cm) was observed in the coffee (*C. arabica* var. *typica*) over the course
of the near normal and the more pronounced dry seasons studied. These findings suggest a spatial and
temporal partitioning of soil water sources between shade trees and coffee plants during drier periods
and water-resource complementary in this coexistence species environment.

Although comparisons of our findings with other traditional shade Arabica coffee plantations

are difficult because studies are essentially lacking in this type of agroecosystems, there are a handful
of other investigations carried out in shade coffee monospecific plantations in the humid tropics in
which complementary rather than competitive water use strategies prevailed. For example, Cannavo
et al. (2011) compared the water use and soil water availability of an unshaded coffee vs. a shaded
monoculture (*Inga densiflora*) coffee plantation in Costa Rica, both of 7-8 years old, using soil
moisture measurements and water balance calculations. Their results showed that soil water content
in the deeper soil layers (> 120 cm depth) was lower in the shaded coffee than in the sun-grown coffee
system, while water content in the shallower layers was similar. This suggested that associated shade
trees   preferentially   used   water   from   deeper   soil   horizons   providing   some   evidence   of
complementarity water use between coffee plants and native *Inga* trees during the dry season.
However, the authors acknowledged that they were unable to separate roots of coffee from those of
trees in the soil profiles, so they could not be certain whether trees were the only individuals extracting
water from deeper sources. In this respect, our study showed that there was always a mixture in water
uptake from different sources (soil group depths), but a separation between the main sources of water
for shade trees and coffee shrubs clearly prevailed.

Other investigations in Costa Rica have examined the belowground resource competition of

Arabica coffee in association with fast-growing timber species using data of plant growth, root
distribution and density, and soil moisture and nutrients patterns. For example, the study of Schaller
et al. (2003) carried out in a commercial (*Eucalyptus deplupta*) shade coffee plantation where soils
are highly fertilized, showed that coffee had a relatively even root distribution along the first 40 cm
of soil depth with a higher root density in the proximity of the coffee rows. Conversely, the root
system of *E. deplupta* was much shallower having most roots concentrated in the upper 10 cm of soil.
In this case, the tree root density was found highest in the alleys between the coffee rows. The authors
explained that the apparent complementary resource exploitation of this tree-crop system was mainly
attributed to high availability of soil resources and the high competitiveness of the coffee limiting the
expansion of tree roots (cf. Lehmann, 2003). Although in our study we did not determine the depth
distribution of coffee and tree roots, our findings showed that all shade tree species were tapping
water from deeper soil layers than coffee, suggesting that trees are deep rooted and being able to
explore larger soil volumes causing little competition with coffee.
In Nicaragua, Padovan et al. (2015) compared the root distribution, soil moisture,
transpiration and leaf water potential patterns in a sun-grown coffee system and an agroforestry of
coffee planted with two timber trees (deciduous *Tabebuia rosea* and evergreen *Simarouba glauca*).
Their findings showed that coffee roots were more abundant than tree roots and mainly concentrated
in the shallower soil layers (0–80 cm depth). Most roots of both tree species were observed in deeper
layers (>100 cm) suggesting a clear niche differentiation with coffee. During the 3-year study period,
volumetric water content along a 2 m soil profile was higher in the sun-grown coffee than in the
shaded coffee, which was explained by greater soil water uptake from trees below the crop rooting
zone (Padovan et al., 2015). Moreover, coffee shrubs in the shaded plantation were more water
stressed (i.e. lowest midday leaf water potentials) during the pronounced dry season studied (Padovan
et al., 2018). Their results suggest that despite the clear hydrological niche segregation, competition
between coffee and shade trees may occur if the dry season is long and severe enough.
Our findings also showed that during the wet season coffee plants substantially increased the
use of near surface water (+56%) in comparison to the dry season, while all shade trees also extended
their water acquisition to much shallower soil water pools (+19%). This is largely explained by the
increases in soil moisture in the first 30 cm depth due to frequent rainfall inputs that characterize the
wet season in our study area. This also suggests that coffee had a higher root activity in the top soil
layers during the wet season in comparison to the dry season, as has been documented in other studies
(Huxley et al., 1974). Regarding the shade trees, we observed that *T. micrantha* showed the greatest
response to wetter conditions by drawing most water from the first 15 cm of soil (92%), whereas this
was much less evident in *L. guatemalensis* (21%) and *I. vera* (27%). Although we did not determine
the vertical distribution of roots for each of the shade tree species studied, these findings suggest that
*T. micrantha* has a shallower rooting system than the other tree species. The fact that the *T. micrantha*
trees were more recently planted (i.e. younger with less developed root system) than the *L.*
*guatemalensis* and *I. vera* trees supports this idea. On other hand, the high temperature and rainfall
that characterize the wet season at our study site may favor rapid mineralization of nutrients and their
subsequent leaching to deeper soil layers (i.e. potassium, calcium and magnesium; Table 4). Hence,
for the larger trees studied (*L. guatemalensis*), the availability of water and nutrients at deeper depths
could have been an important resource for plant growth in this period, partly explaining the lower
activity of their shallower roots. Despite the changes and the higher variability in depth of water
uptake observed among canopy trees and coffee shrubs, a complementary use of soil water prevailed
during the wet season. Future work should be focused on the distribution and dynamics of tree and
crop roots and their seasonal variation in relation to the availability of nutrients and water in the soil.
Also, it would be desirable to relate these dynamics to crop and shade tree phenology to elucidate
temporal synergistic or competitive water requirements.

### 633 *4.3 The role of antecedent wetness in coffee water uptake*

Despite the relatively small sample size, our study showed that antecedent wetness strongly
influenced the water uptake patterns of coffee plants (cf. Huxley et al., 1974). We found that under
relatively wet antecedent conditions prevailing after small rainfall events during the dry season, coffee
substantially increased the use of near surface soil water sources, possibly as an opportunistic strategy
to overcome the soil water deficits in this period and taking advantage of their much shallower rooting
system compared to trees. Conversely, tree water uptake was mainly sourced by deeper soil water
layers showing less sensitiveness to higher antecedent wetness. In this respect there are no
comparative studies in shade coffee agroecosystems evaluating the functional response of plant water
uptake over a range of antecedent wetness. Nevertheless, plant and soil water interactions under dry
and relatively wet conditions have been examined in other types of agroforestry systems. For
example, in the study of Gao et al. (2018) carried out in a semiarid region in China, the authors
evaluated the seasonal variations in water use of jujube (*Ziziphus jujuba*) trees planted with annual
(*Brassica napus*) and perennial (*Hemerocallis fulva*) crops. Using stable isotope techniques and
Bayesian mixing modelling, their results showed that jujube trees generally tapped water (> 58%)
from deep soil layers (60-200 cm depth) at low antecedent wetness, while *B. napus* and *H. fulva* crops
primarily extracted water (> 65%) from intermediate (20-60 cm) and shallow (0-20 cm) soil layers.
This exhibits a complementary water use strategy between trees and crops. However, at higher
antecedent wetness both the jujube trees and crops extracted most water from the first 0-60 cm of soil
depth (> 65%). This indicated that both species exhibited an opportunistic strategy for accessing
resources at shallower soil depths. In this case, contrary to our findings, tree roots rather than crop
roots showed the stronger capacity to switch rapidly from deep to shallow sources in response to
increased soil water availability.

### 657 *4.4 Implications and future directions*

The consistent complementarity in plant water use strategies observed under different
hydrometeorological conditions in the coffee plantation studied provides support to the central tenet
of agroforestry systems (Cannel et al., 1996). Based on our findings, *L. guatemalensis*, *I. vera* and *T.*
*micrantha* provide good choices for coffee shade trees due to their complementarity in soil water use.
Since these tree species obtained their water from deeper soil layers than the coffee, this could mean
that they utilize nutrients leaching beyond the reach of the coffee plants, and so contribute to improved
nutrient cycling and increased overall productivity of the system (van Noordwijk et al., 2015).
Nevertheless, the current outcome may change given the new coffee management practices
that consist on replacing traditional coffee varieties (e.g. *C. arabica* var. *typica*) with others (*C.*
*arabica* var. *costa rica; C. canephora*) that may exhibit deeper roots systems and perhaps different
water (and nutrient) uptake strategies, in response to prevalent diseases such as leaf rust or root
nematodes. Therefore, future research should be focused on evaluating the water source partitioning
of traditional vs. new coffee disease-resistant varieties and their relation to shade tree water use. In
this respect, there are further questions with regard to strategic use of shade tree species, whereby
fast-growing species might be more (commercially) productive but also more competitive. Some
evidence from elsewhere has shown that such management practices do not necessarily increase
competition and may even enhance the water use efficiency as part of drought-avoidance
mechanisms. For example, in southeast China, Wu et al. (2016) used $\delta^2H$ and $\delta^{18}O$ stable isotope
methods to examine the seasonal water use of a fast-growing rubber tree species (*Hevea brasilensis*)
planted with Arabica coffee. Their findings showed that rubber trees were mostly accessing water
from intermediate (15-50 cm depth) and deep soil layers (50-110 cm), meanwhile coffee was mostly
tapping water from the topsoil (< 15 cm). Additionally, rubber trees showed strong root plasticity in
soil water uptake avoiding competition with coffee during the rainy and relatively dry seasons.
However, more research is needed since these results depend largely on tree-crop specie combinations
and local climatic and soil conditions.
In addition to effects of changing management practices, climate warming may induce
changes in plant transpiration throughout the year (e.g. Karmalkar et al., 2011). In our study, we used
a water stable isotope approach along with root and soil macronutrients data to estimate the relatively
contribution of the plant water sources. However, for a more complete assessment of the plant and
soil interactions, seasonal plant water fluxes need to be quantified. Our results so far have made the
first steps towards serving coffee producers to make better decisions on sustainable coffee and water
management, as well as providing new insights into water resources in general, which are urgently
required for implementing efficient and equitable management programs in humid tropical
environments (Hamel et al., 2018). However, future work should be focused on water use of
individual trees and coffee shrubs using ecophysiological and hydrological techniques in order to
know how much water is used from the different soil water pools.

*5. Conclusions*

This study provides the first baseline information on plant water sources for a traditional

shade coffee plantation in the humid tropics. Our results showed that coffee water uptake was mainly
sustained from shallow soil sources (< 15 cm depth) while all shade trees relied on water sources
from deeper soil layers (>15 to 120 cm depth). This complementary strategy in soil water use between
crops and trees was consistent over the course of the near normal and the more pronounced dry
seasons investigated. Across these same periods, we observed that antecedent precipitation had a
strong influence in coffee plants increasing their water uptake to near surface soil water sources as an
opportunistic strategy to overcome the reduced water availability. In the wet season, coffee plants
substantially increased the use of near surface water (< 5 cm depth), whereas shade trees expanded
their water acquisition to the first 15 cm of soil depth. Overall, a greater soil water partitioning
prevailed among tree and coffee species when higher soil moisture conditions were present.
Nevertheless, despite such variability in plant-soil water interactions across seasons, a clear spatial
segregation of the main water source prevailed between shade trees and coffee plants during the rainy
and dry periods investigated.

**Author contributions.** LEMV designed the experiment. LEMV, MSAB and FH collected the field
data. MSAB performed all the Bayesian mixing model analysis. JG contributed in the data analysis.
LEMV prepared the first draft of the manuscript. FH, MSAB and JG edited and commented on the
manuscript several times, and TED carried out the final revision. Later, all the co-authors
contributed with revisions.

**Competing interests.** The authors declare that they have no conflict of interest.

**Acknowledgments**

We would like to thank Raul Monge and Daniel Tejeda for their permission to conduct this

research in "La Orduña" coffee plantation. We also thank Melissa López-Portillo for performing the
cryogenic extractions and Stefania Mambelli and Wembo Yang for analyzing the water isotope
samples at UC-Berkeley, USA. Angel Zaragoza, Erika Mendoza, Alitzel Guzmán and Carlos Alcocer
are thanked for their assistance in the sampling campaigns. The Instituto de Ecología, A.C. (INECOL)
is thanked for the laboratory facilities to carry out the root separation and oven-dry weights. We also
thanked Adriana Hernández and Esperanza Huerta for helping in the root separation process. This
research was supported by the PAPIIT-UNAM (No. IB100313 and IB100113) grants, respectively to
LE Muñoz-Villers and F Holwerda, and by the INFRA-CONACyT-México (No. 187646) grant to F
Holwerda. Additional support was provided by the NSF-USA (No. 1313804) and by the Scottish
Funding Council via the University of Aberdeen (SP10192). Data analysis and partial writing of the
manuscript was performed during LEMV sabbatical leave (March-July 2018) hosted by Josie Geris
at the University of Aberdeen, UK, and granted by the *Programa de Apoyos para la Superación de*
*Personal Académico* (PASPA) of the UNAM. Lastly, we appreciate the valuable comments of
reviewers Adriá Barbeta, Matthias Beyer and Daniele Penna and Editor Matthias Sprenger that helped
to improve an earlier version of the manuscript.

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

**Table 1.** Characteristics of the shade trees and coffee plants sampled for water isotope analysis during 2014 and 2017. Numbers between parentheses are the standard deviation.

| Family | Species | Canopy layer | 2014 | | 2017 | | *n* |
|--------|---------|--------------|------|------|------|------|-----|
| | | | DBH cm | Height m | DBH cm | Height M | |
| Fabaceae | *Lonchocarpus guatemalensis* | Overstory | 101.5 (12.6) | 20.3 (1.3) | 119.8 (12.1) | 21.0 (1.2) | 3 |
| Fabaceae | *Inga vera* | Overstory | 39.3 (15.7) | 10.7 (4.8) | 48.1 (13.3) | 9.6 (1.2) | 3 |
| Cannabaceae | *Trema micrantha* | Overstory | 13.16 (6.8) | 8.15 (3.1) | 23.3 (7.2) | 15.2 (2.2) | 3 |
| Rubiaceae | *Coffea arabica* var. *typica* | Understory | 12.7 (2.1) | 2.83 (0.7) | n.a. | n.a. | 5* 6** 10*** |

* Number of individuals sampled each time in the 2014 dry season

** Number of individuals sampled each time in the 2017 dry season

*** Number of individuals sampled in the 2017 wet season

**Table 2.** Mean ± (SD) H and O stable isotope composition of 2013-2014 precipitation, tree xylem water and bulk soil water of the 2014 dry season sampling, and corresponding *d*-excess values (‰)

| Precipitation $n = 41$ | | | | | | Bulk soil water $n = 54$ | | | | | | | | | | | | Shade trees xylem water $n = 27$ | | | Coffee shrubs xylem water $n = 14$ | | |
|---|---|---|---|---|---|---|---|---|---|---|---|---|---|---|---|---|---|---|---|---|---|---|---|
| Dry season | | | Wet season | | | 0-5 cm depth | | | >5-15 cm depth | | | >15-30 cm depth | | | >30-120 cm depth | | | | | | | | |
| $\delta^2H$ | $\delta^{18}O$ | *d*-excess | $\delta^2H$ | $\delta^{18}O$ | *d*-excess | $\delta^2H$ | $\delta^{18}O$ | *d*-excess | $\delta^2H$ | $\delta^{18}O$ | *d*-excess | $\delta^2H$ | $\delta^{18}O$ | *d*-excess | $\delta^2H$ | $\delta^{18}O$ | *d*-excess | $\delta^2H$ | $\delta^{18}O$ | *d*-excess | $\delta^2H$ | $\delta^{18}O$ | *d*-excess |
| 1.6 ± 8.5 | −1.9 ± 1.4 | 17.0 ± 5.1 | −42.4 ± 36.1 | −7.2 ± 4.3 | 14.9 ± 2.8 | −20.5 ± 7.8 | −2.4 ± 1.0 | −1.5 ± 4.1 | −30.8 ± 9.4 | −3.7 ± 1.1 | −1.2 ± 6.3 | −54.7 ± 10.3 | −7.0 ± 0.9 | 1.2 ± 6.6 | −66.8 ± 8.6 | −8.7 ± 1.3 | 3.0 ± 4.7 | −55.4 ± 7.6 | −6.2 ± 1.0 | −5.8 ± 4.1 | −25.5 ± 10.8 | −3.4 ± 1.8 | 1.7 ± 5.0 |


**Table 3.** Mean ± (SD) H and O stable isotope composition of 2016-2017 precipitation, tree xylem water and bulk soil water of 2017 dry season sampling, and corresponding *d*-excess values (‰)

| Precipitation n = 39 | | | | | | Bulk soil water n = 54 | | | | | | | | | | | | Shade trees xylem water n = 24 | | | Coffee shrubs xylem water n = 18 | | |
|---|---|---|---|---|---|---|---|---|---|---|---|---|---|---|---|---|---|---|---|---|---|---|---|
| Dry season | | | Wet season | | | 0-5 cm depth | | | >5-15 cm depth | | | >15-30 cm depth | | | >30-120 cm depth | | | | | | | | |
| $\delta^2$H | $\delta^{18}$O | *d*-excess | $\delta^2$H | $\delta^{18}$O | *d*-excess | $\delta^2$H | $\delta^{18}$O | *d*-excess | $\delta^2$H | $\delta^{18}$O | *d*-excess | $\delta^2$H | $\delta^{18}$O | *d*-excess | $\delta^2$H | $\delta^{18}$O | *d*-excess | $\delta^2$H | $\delta^{18}$O | *d*-excess | $\delta^2$H | $\delta^{18}$O | *d*-excess |
| −2.9 ± 16.0 | −3.0 ± 1.8 | 21.5 ± 4.3 | −47.8 ± 34.4 | −7.9 ± 4.1 | 15.2 ± 3.3 | −24.3 ± 3.9 | −2.2 ± 0.5 | −6.9 ± 6.6 | −32.1 ± 5.3 | −3.6 ± 0.5 | −3.4 ± 4.1 | −41.9 ± 5.7 | −5.7 ± 0.6 | 3.4 ± 4.8 | −47.3 ± 3.8 | −6.5 ± 0.5 | 5.0 ± 3.2 | −44.9 ± 5.6 | −4.4 ± 0.7 | −9.7 ± 5.4 | −21.3 ± 7.2 | −2.8 ± 1.0 | 1.3 ± 6.2 |

**Table 4.** Soil characteristics (average values) determined at the different depths

| Soil depth | pH (H$_2$O) | P | Na | K | Ca | Mg | CEC | BS | OM | C | N | Clay | Loam | Sand |
|---|---|---|---|---|---|---|---|---|---|---|---|---|---|---|
| (cm) | | (mg kg$^{-1}$) | (cmol$_c$ kg$^{-1}$) | | | | | (%) | | | | | | |
| 5 | 4.07 | 33.33 | 1.47 | 0.60 | 3.86 | 0.87 | 16.10 | 0.42 | 5.18 | 3.01 | 0.38 | 60.83 | 25.1 | 13.9 |
| 15 | 4.12 | 4.60 | 1.08 | 0.47 | 0.95 | 0.12 | 13.27 | 0.20 | 2.89 | 1.90 | 0.30 | 63.8 | 24.3 | 11.9 |
| 30 | 4.34 | n.d. | 2.22 | 0.77 | 1.92 | 0.54 | 14.65 | 0.37 | 1.55 | 1.31 | 0.23 | 70.9 | 18.6 | 10.5 |
| 60 | 4.95 | n.d. | 2.36 | 0.93 | 3.81 | 1.21 | 20.35 | 0.41 | 1.02 | 0.69 | 0.22 | 66.9 | 16.3 | 16.8 |
| 90 | 5.10 | n.d. | 2.75 | 1.11 | 3.78 | 1.27 | 18.85 | 0.47 | 0.48 | 0.50 | 0.20 | 66.1 | 14.9 | 19.1 |
| 120 | 5.16 | n.d. | 3.00 | 1.45 | 3.76 | 1.20 | 17.60 | 0.53 | 0.41 | 0.51 | 0.20 | 65.1 | 14.0 | 20.9 |

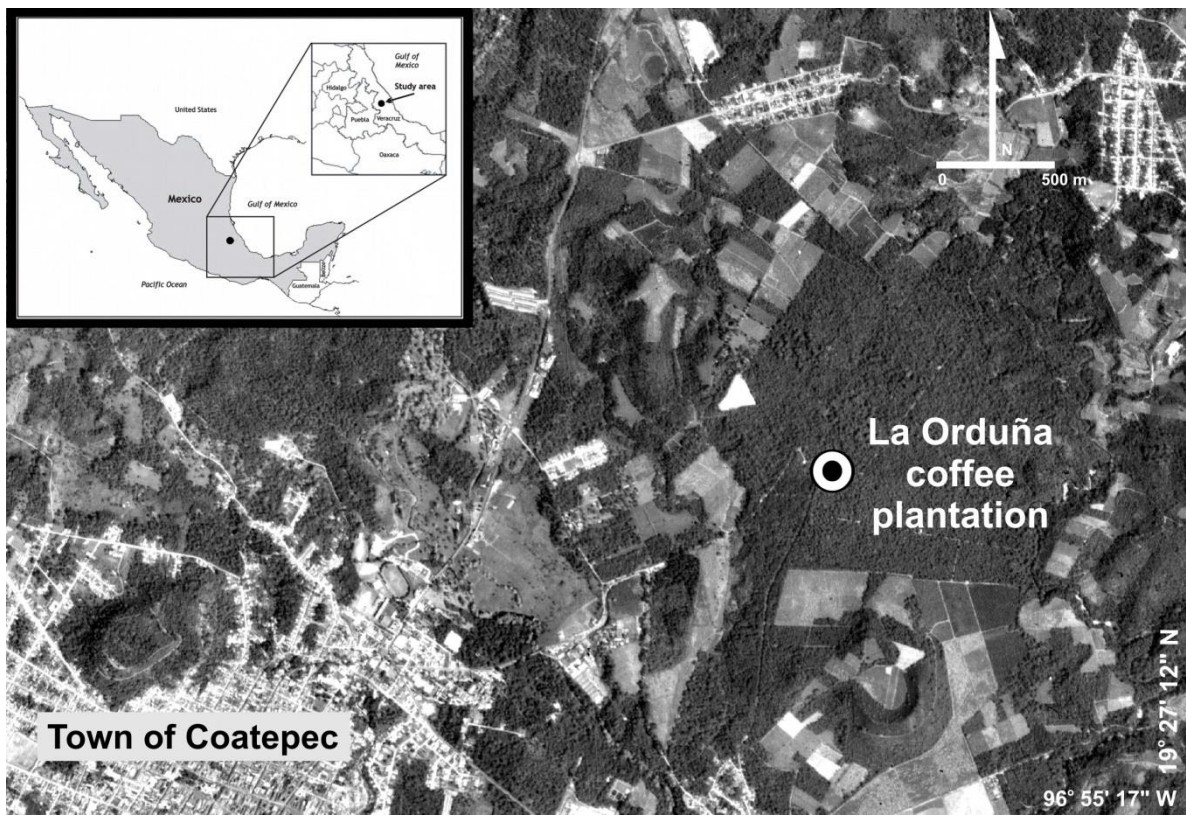

**Figure 1.** Study site location in the municipality of Coatepec, Veracruz, Mexico. Source: QuickBird Satellite Image (2010). Copyright DigitalGlobe, Inc.

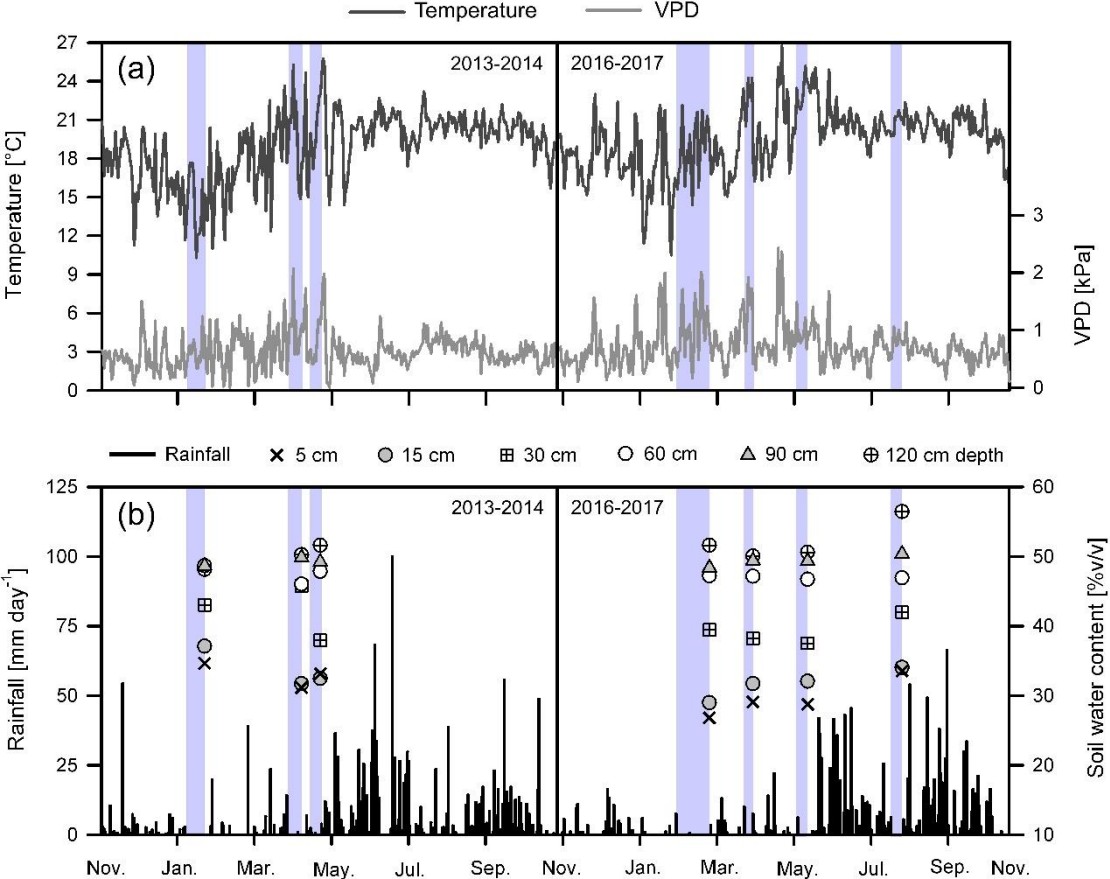

**Figure 2.** (a) Daily mean air temperature and vapor pressure deficit (VPD) and (b) and daily total rainfall (*P*), as measured from November 2013 to October 2014 and from November 2016 to October 2017, and volumetric soil water content (SWC) measured at different depths during the sampling campaigns in the study area; different depths are indicated by the unique symbols shown in the lower panels (the key to the symbols is at top). The blue-colored areas indicate the 6- to 22-day period of minimum rainfall (< 5 mm) preceding the dates of isotope sampling in January (mid dry season) and April (late dry season) of 2014, and in February (mid dry season), April and May (late and end of dry season), and August (mid wet season) of 2017.

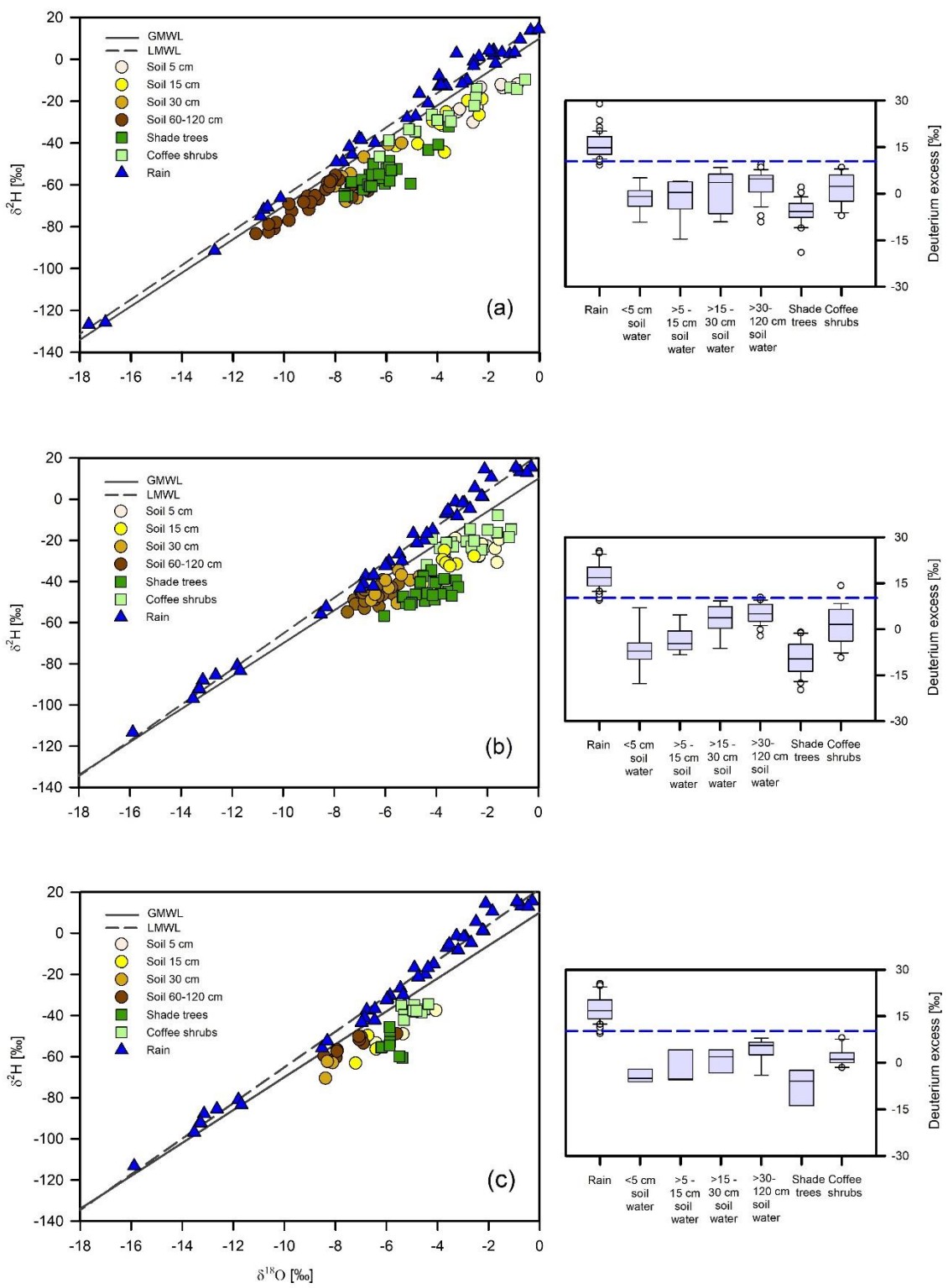

**Figure 3.** (a) Isotope composition of xylem water for shade trees and coffee shrubs, bulk soil at different depths as observed during the three sampling dates (Jan. 23, Apr. 11 and Apr. 26, 2014),

and isotope values of rainfall during the period December 2013 to November 2014. The dashed line represents the 2013–2014 local meteoric water line (LMWL; $\delta^2H = 17.82 + 8.26* \delta^{18}O$), (b) Isotope composition of xylem water for shade trees and coffee shrubs, bulk soil at different depths during the three sampling dates (Feb. 27, Apr. 5 and May. 20, 2017) and isotope values of rainfall during the period December 2016 to November 2017, and (c) Isotope composition of xylem water for shade trees and coffee shrubs, bulk soil at different depths during the middle of the 2017 wet season (Aug. 4) and isotope values of rainfall during the period December 2016 to November 2017. The dashed lines in panels (b) and (c) represent the 2016–2017 local meteoric water line (LMWL; $\delta^2H = 21.0 + 8.36* \delta^{18}O$). The solid line in all panels represents the global meteoric water line (GMWL; $\delta^2H = 10 + 8* \delta^{18}O$). The panels on the right show the deuterium excess values for the plants and soil water sources and rainfall preceding the sampling campaigns. The dashed blue line represents the deuterium excess value of the GMWL.

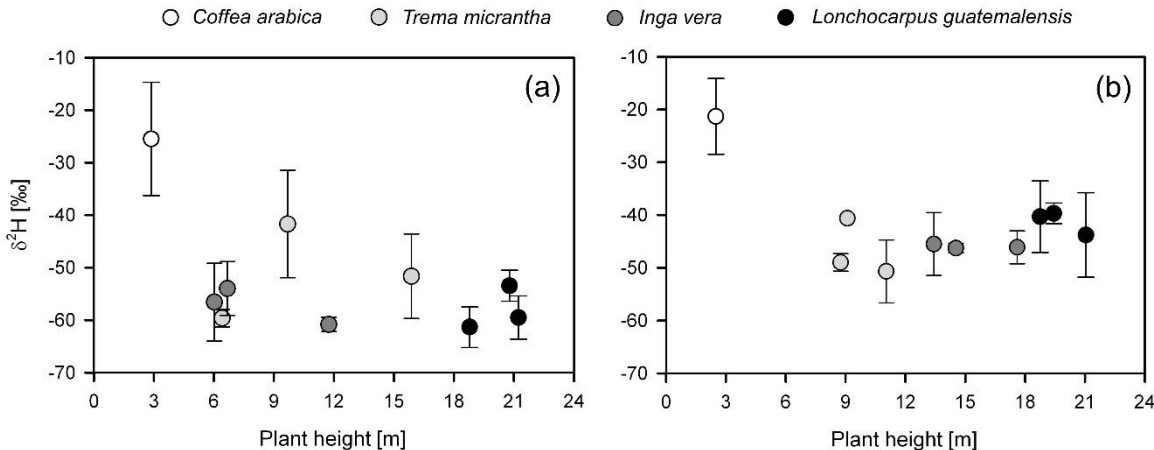

**Figure 4.** Plant height vs $\delta^2$H xylem water for coffee plants and shade tree species corresponding to (a) the 2014 and (b) 2017 dry season samplings.

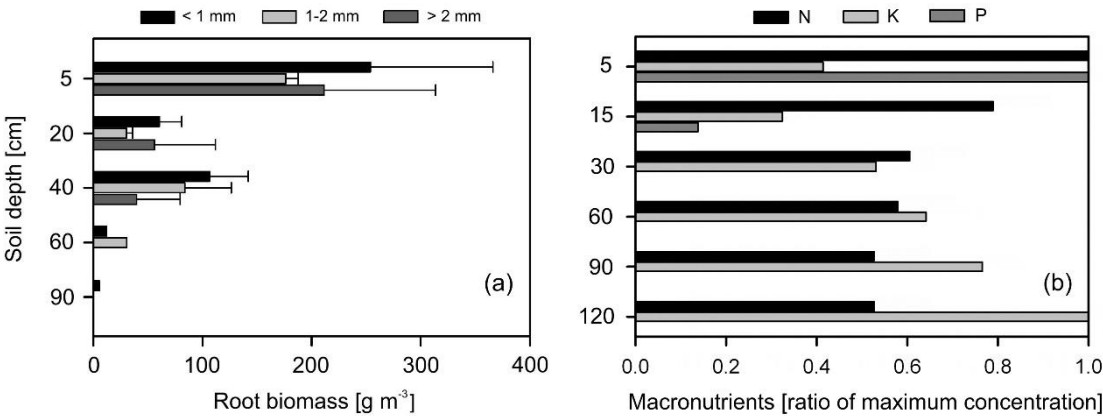

**Figure 5.** (a) Distribution of root biomass for three size classes of roots (different color bars), the error bars in represent one standard deviation of uncertainty and (b) macronutrients distribution along the soil profile, here normalized and expressed as in ratio to their maximum values (absolute values in Table 4).

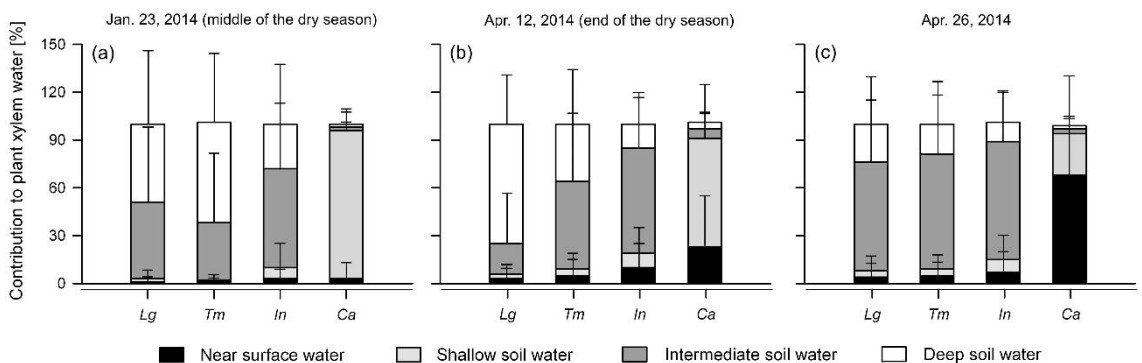

**Figure 6.** MixSIAR Bayesian mixing model results showing the mean likely contribution of each water source to the xylem water of shade canopy trees and coffee shrubs. (a), (b) and (c) show results for the sampling dates of Jan. 23, Apr. 12 and Apr. 26, 2014 respectively, using the informative prior distribution. *Lg: L. guatemalensis; Tm: T. micrantha; In: I. vera and Ca: Coffea arabica*. Error bars represent one standard deviation of uncertainty.

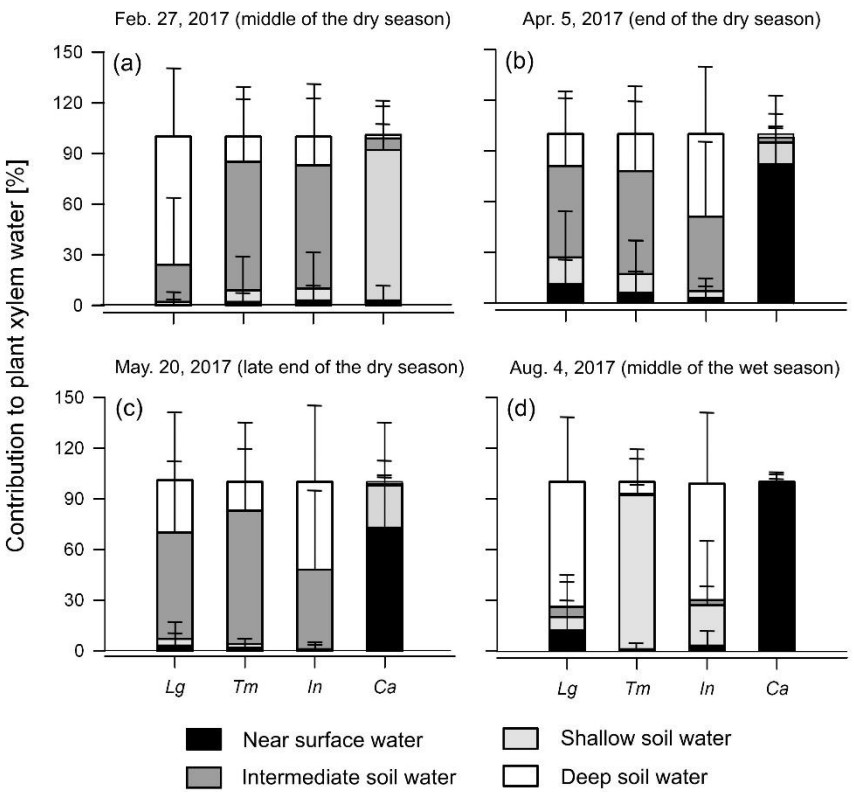

**Figure 7.** MixSIAR Bayesian mixing model results showing the mean likely contribution of each water source to the xylem water of shade canopy trees and coffee shrubs. (a), (b), (c) and (d) show results for the sampling dates of Feb. 27, Apr. 5, May. 20 and Aug. 4, 2017 respectively, using the informative prior distribution. *Lg: L. guatemalensis; Tm: T. micrantha; In: I. vera and Ca: Coffea arabica*. Error bars represent one standard deviation of uncertainty.

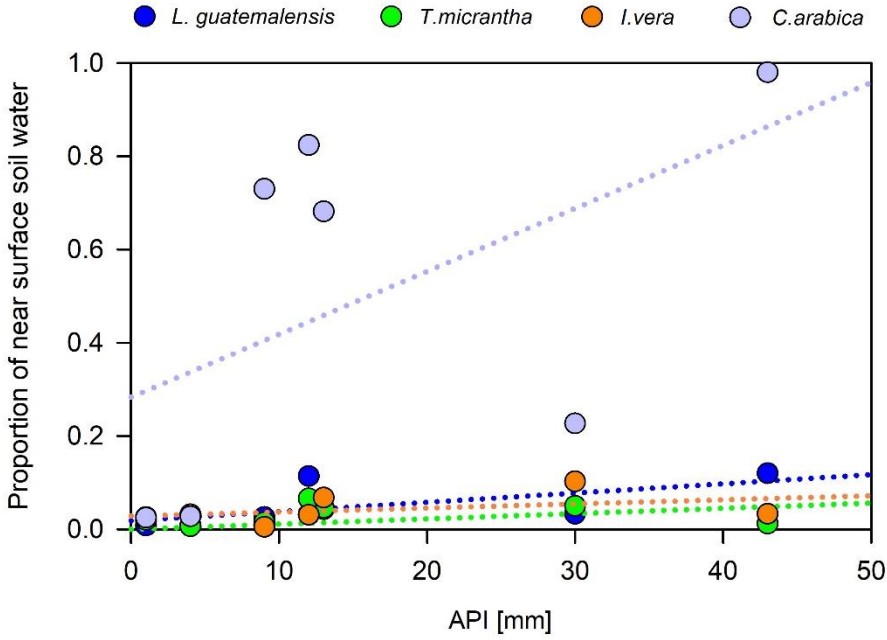

**Figure 8.** Contribution of deep soil water to plant uptake at different antecedent precipitation conditions across the 2014 and 2017 dry seasons.