# Peer review of "Manuscript title"

_Hydrology and Earth System Sciences, 2019_

## Referee Comment (RC1) · Matthias Beyer (Referee) · 15 Aug 2019

Thank you for letting me review the manuscript hess-2019-329 'Coffee and shade trees show complementary use of soil water in a traditional agroforestry ecosystem' by Muñoz-Villers et al. I enjoyed reading. In their work, the authors investigate water uptake depths of large shade trees and coffee trees during two dry seasons and one wet season using water stable isotopes and a Bayesian mixing model. They find that coffee and shade trees show complementary water use patterns, i.e. preferential water use by coffee and deep-water use by the shade trees. During the wet season, both groups shift to shallower resources. Without doubt, this manuscript is well-prepared

and written. The structure is clear, research questions are stated concisely, and the Introduction provides a thorough overview on the topic. The graphics are suitable. I like the study and the topic is interesting. Also – and this is the main scientific contribution of the paper – it is great to see that the authors integrated priors and used root information and macronutrient distributions for that. However, apart from that, the novelty and innovation of the study is limited. I also have a couple of rather major concerns about the methods used in the paper related to the soil water extraction and mixing model. The former might be answered, but the latter might require some more effort. I elaborate on those below. Minor comments are summarized further down.

Good luck and all the best, Matthias Beyer

Major points: l.249-250: was complete extraction somehow validated? Also note that clay-rich soils need higher extraction temperatures (see recent (Gaj et al., 2017; Orlowski et al., 2016) papers on mineral mediated isotope fractionation). Using a water bath at 100°C might result in an offset in isotope compositions and lead to errors/uncertainty in the mixing model (the reservoir of water that is extracted would not equal the reservoir that is available to plants). The authors state at one point that there was an offset of the values towards more depleted – this is exactly what would happen and was observed in other studies when clay was an issue. This issue should be at least discussed. Another question (but this is more general) related to the cryogenic extraction is why such long extraction times are needed (I know, West et al. 2006 propose that). I think one part of that is related to the relatively low extraction temperature, but still. The extractable water should be leaving the sample side very fast given the low volumes (even under 100 °C) – waiting longer would not evaporate more water from the sample side unless the temperature is increased further. l. 297-300: These assumptions need to be validated/proven. Why was not the soil water isotope composition of the first 5 cm used directly? I guess in order to account for water that was taken up by the plant before the actual sampling date? How was the classification used for the mixing model decided? Slightly above and below the zero-flux plane,

the isotope composition of soils normally changes drastically during dry periods. . .for clay this is often in the first 15 cm soil depth. The 30 -120 cm depth were isotopically similar? In my understanding, the discretization used in the mixing model should be done after the isotope depth profiles are evaluated and backed up by statistical measures of differences between different depths. After checking the supplementary data, I'm really doubting the discretization used. There are partially huge differences of the isotope values of the soil profiles between 30 and 120 cm. And how about 15-30 cm? – was the isotope information of this depth not used at all? (in that case, the mixing model is missing a source which violates the mixing model requirements). I refer, once again, to the Rothfuss et al. publication, which might help to address these issues. Minor points: - Since many different analysis were carried out with the soil and plant samples, this could be summarized in a table nicely. - It would have been easy and interesting to check the uptake depths of the large trees separately and not lumping them. (but maybe not of interest for the study) - I suggest strong discussion of the use of informative priors and putting a more general focus on this aspect, as this is the key scientific/methodological novelty in this paper in my opinion. - (more a comment): It would have been interesting to have water potential measurements in both soils and trees, because those could really constrain the possible uptake depths.

Abstract l.27: Providing the rainfall amounts in addition to the year would be nice; in addition, it would be nice if the authors could state the type of environment of the study (e.g. semi-arid, tropical,. . .) ll.35/36: the percentages are the mean? median? I suggest adding a +/- xx % notation accounting for uncertainty l.39: short-term wetness status? Do the authors mean that the uptake depth is not influenced by small rain events? This sentence is not easy to understand, I suggest rephrasing ll.39-41: this sentence needs to be rephrased. The terms near surface vs. much shallower are confusing the reader (5 and 15 cm are both shallow). Perhaps 'upper five centimeter'? ll.42-43: the spatial segregation mentioned, is it due to the different rooting depths of the studied plants? Was this validated somehow? l.44: plant-soil water uptake? Confusing phrase. Do the authors mean 'root water uptake patterns/depths'? I feel like a

concluding sentence is missing in the abstract. What are the implications of the study? What novel things were found out? Is 120 cm the max. rooting depth??? Uptake depth vs. rooting depth? (coffee shallow, others deep) Introduction I really like the way the introduction is written (clear and concise). The Bayesian mixing model needs to be addressed though. The word is only mentioned once, and some readers might not know what it even is. At the end of the introduction, sentence is missing highlighting the importance and novelty of this research. l.55 and l.73: 'soil resources' sounds odd. . .can the authors specify please? l.87: However, l. 90-92: please note that mixing models are also frequently criticized, (Rothfuss and Javaux, 2016) l.92: 'Although rarely implemented' – do the authors have examples where it was implemented? (this is out of interest) l.143: micrometeorological measurements (which) l. 146: nice the authors are implementing priors. See related publication where this was suggested (and also MixSIAR was used): (Beyer et al., 2018). You don't have to cite us but maybe it helps for some explanation in the authors manuscript. l.151/152: The answer to question no. 2 is not reflected in the abstract Materials/Methods l. 168: on an; is there no data after 2000 for rainfall? This seems like it's likely to have changed meanwhile l.214: 'carried out' rather than 'performed'? l.218-222: how many replicates per individual were taken? (same later for coffee and the soil samples) l.232-233: 'Auger sampling points were located so that each of the sampled shade trees and coffee plants had a total of three soil sampling points within their 3 m radius.' – If it was sampled at only three different locations (see sentence before), so it means that all the trees had the three sampling points in their 3m radius? That seems odd. Can the authors please check if this phrasing is correct here? l.247: refrigerated – was any mold developing on the samples? This can affect isotope ratios l.268/269: What is API – if it is not a common method, it needs to be explained briefly. l.304/305: It would be very appreciable to the community I believe if the authors explain how the priors were determined and implemented into MixSIAR as this is not something that has been done often. Results l.321: I see a point in putting this as result, but this is nothing that belongs to the objectives of the study as such. I suggest including it into the methods chapter. In

many hydrologic and soil studies variables such as rainfall and soil moisture are the basis and not highlighted as results. l. 335: Definition of normal vs. below-average dry season: In fact, both dry seasons sampled were below average, 2014 was about 20% lower (323 mm vs. 389 mm normal) and the 2016/17 one 40%....not sure if I would consider 20% below average a 'normal' year. l.351-353: it is not surprising that the wet season is wetter the dry season, but it is notable that the wet season is drier than the 2014 dry season! Why is this information omitted? l.353: the API results don't tell the reader anything without proper explanation ll.359-360: two digits after comma reported for 18O – more than precision – should be avoided; add 'for' delta 18O, 'for' delta 2H l.382-384: because of the effect of clay material on extraction? (see comment before) – same for ll. 387-388 l.417: the root biomass cannot be distinguished between species, right? (coffee vs. large trees?). . .that means that the created informative prior would be quite biased. . .. ll.432-436: discussion Putting the rainfall amounts in the results section is debatable. . .it sure is something that was done during the study, but it is not directly related to the objectives. As Hydrologist, I personally would've liked to read these numbers earlier to put the words 'dry season', 'less than average' etc. in perspective. Discussion Ll.522-525: So in the wet season both trees and coffee use shallow water, because it's abundant. In the dry season, the trees use deep water – because they have deeper roots and water in deeper soil is easier accessible (low matric potential of soils). The coffee uses shallower water in the dry season. What is the reason? – the fact that coffee plants cannot grow deep roots? – or is it because they don't need so much water compared to the trees and don't need deep roots? – or, because the coffee plant has another strategy and its roots can extract water from drier soil compared to tree roots? or. . ..... This is not a criticism; this question is out of interest. I wonder then, if this is really 'complementary' water use as such? ll.599-600: Which recommendations based on their results would the authors give to coffee producers then? This would be a nice addition. ll.606-612: this is a bit contradictory, because in the presented example using this additionally information did not affect the results much (both uncertainty and general outcomes). So which variables should be

included in the future? Are there others that might be more suitable? Micronutrients? Soil moisture?.... Conclusions An experienced and well-known researcher a while ago gave me the advice: 'A good paper does not need a conclusion chapter – the reader draws them him/herself.' That stuck to me somehow. I think this is a good paper.

Orlowski, N., Breuer, L., McDonnell, J.J., 2016. Critical issues with cryogenic extraction of soil water for stable isotope analysis. Ecohydrology 9, 1–5. https://doi.org/10.1002/eco.1722

Rothfuss, Y., Javaux, M., 2016. Isotopic approaches to quantifying root water uptake and redistribution: a review and comparison of methods. Biogeosciences Discuss. 1–47. https://doi.org/10.5194/bg-2016-410

Gaj, M., Kaufhold, S., Koeniger, P., Beyer, M., Weiler, M., Himmelsbach, T., 2017. Mineral mediated isotope fractionation of soil water. Rapid Commun. Mass Spectrom. 31, 269–280. https://doi.org/10.1002/rcm.7787

Beyer, M., Hamutoko, J.T., Wanke, H., Gaj, M., Koeniger, P., 2018. Examination of deep root water uptake using anomalies of soil water stable isotopes, depth-controlled isotopic labeling and mixing models. J. Hydrol. 566, 122–136. https://doi.org/10.1016/J.JHYDROL.2018.08.060

Please also note the supplement to this comment:
https://www.hydrol-earth-syst-sci-discuss.net/hess-2019-329/hess-2019-329-RC1-supplement.pdf

---

## Referee Comment (RC2) · Adrià Barbeta (Referee) · 20 Aug 2019

Review of manuscript HESS-2019-329

General comment

This study analyses plant water source partitioning in a coffee agroforestry system along seasons with contrasting soil moisture conditions. For that, the authors applied stable isotope techniques and Bayesian mixing models (MixSIAR) in order to test for the complementary use of soil water in space and time by coffee plants and shade trees. The importance of the study comes from the fact that ecohydrological relations in this type of traditional agroforestry systems are completely unknown, in contrast to those of intensive monospecific plantations. A novel aspect of the study is the inclusion of root and nutrient distributions within the framework of stable isotope mixing models, which is a usually underestimated capability of such models. That should improve their accuracy since plant water source partitioning is obviously constrained by root distribution and soil profiles of nutrient availability. Overall, this is a well-designed, rigorous study, that is also clearly presented and well-written. Methods and results are concisely described and figures and tables are easy to interpret. Similar studies of plant water source partitioning are numerous, so it could be said that this study is not especially original. However, I find valuable to report this type of data from regions where they are scarce (*i.e.* Central and South America or Africa, see Barbeta & Peñuelas, 2017; Evaristo & Mcdonnell, 2017).

Although my general assessment of the manuscript is highly positive, I miss some caution regarding stable isotope techniques. While this is a well-established approach, recent studies pointed to methodological issues linked to fractionation processes within the soil matrix (Orlowski *et al.*, 2018; Gaj *et al.*, 2019; Oerter & Bowen, 2019; Oerter *et al.*, 2019), along the soil-plant continuum (Vargas *et al.*, 2017; Barbeta *et al.*, 2019) or within plant tissues (Zhao *et al.*, 2016). Not all ecohydrological systems may be affected by those fractionation processes, and oxygen isotopes seem to still be highly reliable (Zhao *et al.*, 2016; Vargas *et al.*, 2017; Barbeta *et al.*, 2019). Still, in Fig. 3, I observe that xylem water isotopes do not match very well with soil water isotopes from either depth. This is clearer for shade trees. A similar pattern arises in the deuterium excess boxplots. A thorough consideration of potential fractionation processes would require extensive additional analyses, which I think that it is not realistic to ask the authors to do. A more plausible solution is an explanation on why the authors think that fractionation processes are not relevant for their study. It might also be considered to run MixSIAR models separately for oxygen and hydrogen isotopes to check if there are significant discrepancies between them (as in Evaristo *et al.*, 2017; Barbeta *et al.*, 2019). As I said, it is known that fractionation processes do not affect in the same proportion oxygen and hydrogen isotopes. In any case, I believe that these emerging issues cannot longer be ignored by plant water source studies using stable isotopes.

Minor comments

L38 It is not completely clear what does 'precipitation conditions' mean.

L65 Species name (Cedrela odorata) should not be in capital letters.

L191 The high clay content is likely to produce soil water isotopic fractionation (Oerter *et al.*, 2014).

L218 The sampling of different plant parts in coffee plants and shade trees (cores VS branches) could have led to a different proportion of internal plant water pools in the xylem water samples of each group.

L223 I assume that bark was peeled off from coffee shrubs, too.

L298 Recent precipitation, especially in periods with relatively wet soil conditions, could in fact percolate faster towards deeper layers. So, rainfall is not necessarily representative of near surface soil water.

L304 The use of prior information is a very interesting point of the study.

References

**Barbeta A, Jones SP, Clavé L, Wingate L, Gimeno TE, Fréjaville B, Wohl S, Ogée J**. **2019**. Unexplained hydrogen isotope offsets complicate the identification and quantification of tree water sources in a riparian forest. *Hydrology and Earth System Sciences Discussions* **23**: 1–31.

**Barbeta A, Peñuelas J**. **2017**. Relative contribution of groundwater to plant transpiration estimated with stable isotopes. *Scientific reports* **7**: 10580.

**Evaristo J, Mcdonnell JJ**. **2017**. Prevalence and magnitude of groundwater use by vegetation : a global stable isotope meta-analysis. *Scientific reports*: 1–12.

**Evaristo J, McDonnell JJ, Clemens J**. **2017**. Plant source water apportionment using stable isotopes: A comparison of simple linear, two-compartment mixing model approaches. *Hydrological Processes* **31**: 3750–3758.

**Gaj M, Lamparter A, Woche SK, Bachmann J, McDonnell JJ, Stange CF**. **2019**. The Role of Matric Potential, Solid Interfacial Chemistry, and Wettability on Isotopic Equilibrium Fractionation. *Vadose Zone Journal* **18**: 0.

**Oerter EJ, Bowen GJ**. **2019**. Spatiotemporal heterogeneity in soil water stable isotopic composition and its ecohydrologic implications in semi-arid ecosystems. *Hydrological Processes*: 0–2.

**Oerter E, Finstad K, Schaefer J, Goldsmith GR, Dawson T, Amundson R**. **2014**. Oxygen isotope fractionation effects in soil water via interaction with cations (Mg, Ca, K, Na) adsorbed to phyllosilicate clay minerals. *Journal of Hydrology* **515**: 1–9.

**Oerter EJ, Siebert G, Bowling DR, Bowen G**. **2019**. Soil water vapor isotopes identify missing water source for streamside trees. *Ecohydrology*: e2083.

**Orlowski N, Breuer L, Angeli N, Boeckx P**. **2018**. Inter-laboratory comparison of cryogenic water extraction systems for stable isotope analysis of soil water. *Hydrology and Earth System Sciences*: 1–36.

**Vargas AI, Schaffer B, Yuhong L, Sternberg L da SL**. **2017**. Testing plant use of mobile vs immobile soil water sources using stable isotope experiments. *New Phytologist* **215**: 582–594.

**Zhao L, Wang L, Cernusak LA, Liu X, Xiao H, Zhou M, Zhang S**. **2016**. Significant Difference in Hydrogen Isotope Composition Between Xylem and Tissue Water in Populus Euphratica. *Plant Cell and Environment* **39**: 1848–1857.

---

## Referee Comment (RC3) · Daniele Penna (Referee) · 31 Aug 2019

Review of the manuscript 'Coffee and shade trees show complementary use of soil water in a traditional agroforestry ecosystem' by Muñoz-Villers et al. (hess-2019-329)

General comment

Reading this manuscript was pleasant. I particularly liked the idea of including the distribution of roots and nutrients in the mixing model approach, and I think that this point should be better stressed in the manuscript.

Overall, I fully agree with the comments provided by the reviews of Matthias Beyer and

[Figure]

Adrià Barbeta, and I have only some minor comments to add.

Good job!

Daniele

Minor comments and technical corrections 42-43. This sentence is not immediate to understand without reading the paper. I suggest rephrasing.

45. Double negation (limitation. . .absent): I suggest simplifying the sentence.

64. Complex sentence, rephrase.

149. I suggest to change into ". . .prevails over competition. . ." or, in any case, to include both the terms "complementary" and "competition" because the latter is logically linked to the second research question.

268. Did you consider using the Normalized Antecedent Precipitation Index (NAPI, Heggen, 2001), instead of API? Heggen, R.J., 2001. Normalized antecedent precipitation index. J. Hydrol. Eng.

385-405. I suggest to condense this part and let the figures talk for themselves.

Fig. 3. Caption: why panel (c) shows the GMWL whereas panels (a) and (b) the LMWL?

Fig. 4. I suggest to replace "(a)" and "(b)" with "2014" and "2017" for more immediate understanding.

Fig. 5. What do error bars represent? Why are there only in panel (a) and not in panel (b)?

Fig. 8. I think that the result and discussion build around this figure should be taken with a bit of caution because based on few point only. I suggest to discuss this limitation in the manuscript.

[Figure]

---

## Author Comment (AC1) · 18 Oct 2019

We thank Matthias Beyer for his positive and constructive comments which allow us to further improve the article. Please find below our response to each of the comments.

[1] Major points: l.249-250: was complete extraction somehow validated? Also note that clay-rich soils need higher extraction temperatures (see recent (Gaj et al., 2017; Orlowski et al., 2016) papers on mineral mediated isotope fractionation). Using a water bath at 100_C might result in an offset in isotope compositions and lead to errors/ uncertainty in the mixing model (the reservoir of water that is extracted would not equal the reservoir that is available to plants). The authors state at one point that there was

an offset of the values towards more depleted – this is exactly what would happen and was observed in other studies when clay was an issue. This issue should be at least discussed.

Reply: Validation of complete extraction. We did not check whether all water was extracted using a gravimetric water content assessment. However, according to the findings of Araguas-Araguas et al. (1995) and West et al. (2006), extractions do not have to reach full completion (i.e., all water extracted) to obtain an unfractionated and, therefore, isotopically consistent value. Experiments have shown that the isotope value of any extracted water increased quickly during the first 20-75 minutes of extraction, after which the isotope value of the extracted water remained constant regardless of further increases in extraction time. The time at which this threshold is reached is the minimum extraction time (Tmin) required to obtain an isotopically unfractionated water sample, and once Tmin is reached, only a very small amount (microliters) of water may remain in the sample. Recently, Orlowski et al. (2013) showed that even if the extraction is conducted until what they claimed was complete, the isotopic signature may not be recovered from different soil types. The Tmin value varied depending on the source material. West et al. (2016) showed that woody stems required the longest extraction times (60–75 min), while values of Tmin were shorter for soil (40 and 30 min for clay and sand soil textures, respectively). Following West et al. (2016), we used the same extraction time for stems and soils (60-70 min).

Clay-rich soils need higher extraction temperatures. Apart from extraction duration, the literature has shown that the extraction temperature "might" have an impact in the soil isotopic composition. Araguas-Araguas et al. (1995) showed that a highly mobile water reservoir that is weakly bound to soil particles can exist (especially in clay-rich soils where interlayered water can be present), and remains largely intact at extraction temperatures < 100°C. More recently, the studies of Orlowski et al. (2016) and Schoonheydt and Johston (2015) have discussed whether the extraction temperature should be increased. However, there has been no systematic investigation that clearly

identified the driving forces that might cause an isotope effect on the isotopic composition of the extracted soil water. Since it has been shown that soil samples containing a high clay fraction might affect the quality of the soil water extraction, and therefore the isotopic composition of the bound water, several papers have suggested that investigations should now incorporate information of the soil hydro-physical properties, and more importantly for clayey soils, information about the cation exchange capacity (CEC), as Vidal and Dubacq (2009) have pointed out that the effect of this interlayered space/water in clay-rich soils can be indirectly evaluated with CEC. For our study, we did determine other soil physical and chemical properties such as CEC. Therefore, we are going to incorporate this information in the manuscript to show that the contribution of this interlayer water bound in the clay mineral structure was small for our soils, and therefore of little significance for the entire isotopic composition of the extracted soil water, and for the mixing model results.

Importantly, we did state that the values of $\delta 2H$ and $\delta 18O$ plant xylem ($-40.8 \pm 15.0‰$ and $-4.6 \pm 1.6‰$ respectively) were on average more positive in comparison to bulk soil water ($-46.7 \pm 16.4‰$ and $-6.0 \pm 2.3‰$ respectively) (L386-388); however, the isotopic range of plant xylem water ($-7.64$ to $-0.56$ for 18O, and $-65.47$ to $-9.64$ for 2H) fell within the bulk soil isotope range ($-11.10$ to $-0.87$ for 18O, and $-83.35$ to $-11.86$ for 2H), and no statistically significant differences were found ($p > 0.05$). Therefore, instead of reporting the mean and standard deviation values, we will present the isotopic signature ranges to show that overall we had a good isotopic match between the soil pore and xylem water.

[2] Another question (but this is more general) related to the cryogenic extraction is why such long extraction times are needed (I know, West et al. 2006 propose that). I think one part of that is related to the relatively low extraction temperature, but still. The extractable water should be leaving the sample side very fast given the low volumes (even under 100 _C) – waiting longer would not evaporate more water from the sample side unless the temperature is increased further.

Reply: Please see our reply to your previous comment.

[3] l. 297-300: These assumptions need to be validated/proven. Why was not the soil water isotope composition of the first 5 cm used directly? I guess in order to account for water that was taken up by the plant before the actual sampling date?

Reply: We have revisited this assumption. Since each isotope sampling campaign was preceded by at least 6 days up to 22 days without or with minimum accumulated rainfall (< 5 mm) (L235-236), we acknowledge the difficulties to consider rainfall as a potential source of near surface soil water. Following the suggestion, we have decided to use the isotopic composition of the soil at 5 cm depth as a source for near surface soil water. Thus, the discretization of the mixing model originally presented is going to be modified in the revised version; the methods and results will reflect this change accordingly. Please also see our reply to the next comment.

[4] How was the classification used for the mixing model decided? Slightly above and below the zero-flux plane, the isotope composition of soils normally changes drastically during dry periods: : :for clay this is often in the first 15 cm soil depth. The 30 -120 cm depth were isotopically similar? In my understanding, the discretization used in the mixing model should be done after the isotope depth profiles are evaluated and backed up by statistical measures of differences between different depths. After checking the supplementary data, I'm really doubting the discretization used. There are partially huge differences of the isotope values of the soil profiles between 30 and 120 cm. And how about 15-30 cm? – was the isotope information of this depth not used at all? (in that case, the mixing model is missing a source which violates the mixing model requirements). I refer, once again, to the Rothfuss et al. publication, which might help to address these issues.

Reply: The classification used in the mixing model was based on the changes in the isotopic composition of soil water and the changes in the root and nutrient distributions along the profile. We divided the soil water pool in two compartments: shallow (5-15

cm depth) and deep soil (30-120 cm depth) sources. In each campaign, we sampled the soils for isotopes at the following depths: 5, 15, 30, 60, 90 and 120 cm. Further, we classified the soil isotope data collected at 5 and 15 cm as shallow and those obtained at 30, 60, 90 and 120 cm depth as deep. Thus, the potential tree water sources that we considered were restricted to these categories and data. There are other examples in the literature in which the evaluation of the relative contribution of soil water sources to plant uptake has been restricted to particular groups of soil depth (cf. Barbeta et al. 2019), without violating the mixing model requirements. However, since the isotopic composition at 5 cm depth is going to be used as the near surface water source, we performed some statistical tests to define the new classification of the soil water pool. Based on the results of these tests, the soil water pool will be divided in the following compartments: near surface (5 cm depth), shallow (15 cm depth), intermediate (30 cm depth) and deep (average of 60-120 cm depth) soil water sources. Preliminary runs of the Bayesian mixing model using this new discretization and without or with the informative prior data continue to show a complementary water use strategy between trees and coffee plants during the dry and wet periods investigated.

[5] Minor points: - Since many different analysis were carried out with the soil and plant samples, this could be summarized in a table nicely. - It would have been easy and interesting to check the uptake depths of the large trees separately and not lumping them. (but maybe not of interest for the study)

Reply: Since these analyses are already described in detail in the text, we consider it redundant to add a table. With regard to the uptake depth, we were unable to distinguish between roots of coffee shrubs and shade trees, as well as between the roots of the different species of shade trees.

[6] - I suggest strong discussion of the use of informative priors and putting a more general focus on this aspect, as this is the key scientific/methodological novelty in this paper in my opinion.

Reply: We will improve this section in the discussion to stress the importance of using informative priors in the mixing models.

[7] - (more a comment): It would have been interesting to have water potential measurements in both soils and trees, because those could really constrain the possible uptake depths.

Reply: Yes, we agree that such data would have been interesting. In a follow-up study, we have been doing water potential measurements at the time of sample collection for isotope analysis.

[8] Abstract l.27: Providing the rainfall amounts in addition to the year would be nice; in addition, it would be nice if the authors could state the type of environment of the study (e.g. semi-arid, tropical,: : :)

Reply: We will add this information.

[9] ll.35/36: the percentages are the mean? median? I suggest adding a +/- xx % notation accounting for uncertainty

Reply: The percentages are mean values; we will add the +/- % standard deviation.

[10] l.39: short-term wetness status? Do the authors mean that the uptake depth is not influenced by small rain events? This sentence is not easy to understand, I suggest rephrasing

Reply: We will rephrase this sentence for clarification.

[11] ll.39-41: this sentence needs to be rephrased. The terms near surface vs. much shallower are confusing the reader (5 and 15 cm are both shallow). Perhaps 'upper five centimeter'?

Reply: We will use the terms mentioned above (i.e., near surface for 5 cm depth and shallow for 15 cm depth).

[12] ll.42-43: the spatial segregation mentioned, is it due to the different rooting depths of the studied plants? Was this validated somehow?

Reply: Please see our reply to a similar previous comment.

[13] l.44: plant-soil water uptake? Confusing phrase. Do the authors mean 'root water uptake patterns/depths'? I feel like a concluding sentence is missing in the abstract. What are the implications of the study? What novel things were found out? Is 120 cm the max. rooting depth??? Uptake depth vs. rooting depth? (coffee shallow, others deep)

Reply: Yes, we mean root water uptake patterns. We will rephrase the sentence. The implications of our study are presented in Section 4.3 (Implications and future direction) in the Discussion. The contribution (novelty) of this research has been argued in the Introduction and the Discussion sections. 120 cm was the deeper potential water source that we examined. Regarding the question about uptake depth vs. rooting depth is hard to answer it in the context of the line 44.

[14] Introduction I really like the way the introduction is written (clear and concise).The Bayesian mixing model needs to be addressed though. The word is only mentioned once, and some readers might not know what it even is. At the end of the introduction, sentence is missing highlighting the importance and novelty of this research.

Reply: We will provide more background information about Bayesian mixing models and highlight the novelty of including priors for the quantification of plant water sources.

[15] l.55 and l.73: 'soil resources' sounds odd: : :can the authors specify please?

Reply: We will specify this.

[16] l.87: However,

Reply: Thank you for the suggestion.

[17] l. 90-92: please note that mixing models are also frequently criticized, (Rothfuss

and Javaux, 2016)

Reply: We are aware that mixing models have been criticized; however, they have several advantages over other methods. That is, they allow for determining the likelihood of the different water sources available to plants using a robust statistical approach and they allow for the incorporation of biophysical parameters (e.g., root and nutrient data) as informative priors (Muñoz-Villers et al. 2018; Ogle et al. 2004).

[18] l.92: 'Although rarely implemented' – do the authors have examples where it was implemented? (this is out of interest)

Reply: To our knowledge, Muñoz-Villers et al. (2018) have been the only ones to use nutrient and root distribution data as priors to better inform a Bayesian mixing model. We will add this reference in the text.

[19] l.143: micrometeorological measurements (which)

Reply: We will change this to "microclimatic measurements". The list of the microclimatic variables that were measured are provided in Section 2.2.

[20] l. 146: nice the authors are implementing priors. See related publication where this was suggested (and also MixSIAR was used): (Beyer et al., 2018). You don't have to cite us but maybe it helps for some explanation in the authors manuscript.

Reply: Thank you for the recommendation.

[21] l.151/152: The answer to question no. 2 is not reflected in the abstract Materials/Methods

Reply: We will rephrase our results in the abstract to include the findings of question #2. With regard to the Materials/Methods, in the Section 2.3 we did say that the dry season of 2017 was warmer and drier offering the opportunity to examine the vegetation responses under more pronounced dry conditions.

[22] l. 168: on an; is there no data after 2000 for rainfall? This seems like it's likely to

have changed meanwhile

Reply: Indeed, there are no data after 2000. And we don't have the data ourselves to determine if there have been any changes in rainfall.

[23] l.214: 'carried out' rather than 'performed'?

Reply: We will make this change.

[24] l.218-222: how many replicates per individual were taken? (same later for coffee and the soil samples)

Reply: This information is given in Tables 1, 2 and 3. For the coffee, the number of replicates is also provided in the text (L225-228). For the trees and soil samples, we can add this information in the text.

[25] l.232-233: 'Auger sampling points were located so that each of the sampled shade trees and coffee plants had a total of three soil sampling points within their 3 m radius.' – If it was sampled at only three different locations (see sentence before), so it means that all the trees had the three sampling points in their 3m radius? That seems odd. Can the authors please check if this phrasing is correct here?

Reply: We will rephrase the sentence for clarification.

[26] l.247: refrigerated – was any mold developing on the samples? This can affect isotope ratios

Reply: Some mold had developed on some of the samples of the trees and coffee shrubs; we will add this in the text and discuss its possible effect in the xylem isotope ratios.

[27] l.268/269: What is API – if it is not a common method, it needs to be explained briefly.

Reply: API stands for antecedent precipitation index and it was calculated following the

method of Viessman et al. (1989) (L267-269). It is actually a common hydrological metric used to quantify the antecedent precipitation conditions (7 or 15 days) prior to a rainfall event, sampling date, etc.

[28] l.304/305: It would be very appreciable to the community I believe if the authors explain how the priors were determined and implemented into MixSIAR as this is not something that has been done often.

Reply: The macronutrients (N, P, K) and root biomass data were first grouped (averaged) according to the defined depths to represent different plant water sources (L306-308). Then, each profile was normalized to obtain a distribution with depth that totalized 100%. Finally, the normalized profiles were averaged across depths to obtain a distribution that represents the prior probability for each source. The prior proportions used for the 2014 sampling dates were: rain = 1, shallow soil = 67, deep soil = 32. For 2017, the following proportions were used: precipitation = 1, shallow soil = 57, deep soil = 42. This configuration resulted in sharp proportions for each source contrasting the "uninformative" prior distribution. We will add this information in the Supplementary Material.

[29] Results l.321: I see a point in putting this as result, but this is nothing that belongs to the objectives of the study as such. I suggest including it into the methods chapter. In many hydrologic and soil studies variables such as rainfall and soil moisture are the basis and not highlighted as results.

Reply: This section characterizes the hydrometeorogical conditions during the two dry seasons (2014 and 2017) and the wet season (2017) studied. Since one of our objectives was to determine the sources of plant water under different soil water availability conditions (L137-144), we consider it important to present this information as part of the Results section.

[30] l. 335: Definition of normal vs. below-average dry season: In fact, both dry seasons sampled were below average, 2014 was about 20% lower (323 mm vs. 389 mm

normal) and the 2016/17 one 40%....not sure if I would consider 20% below average a 'normal' year.

Reply: We will make this more clear.

[31] l.351-353: it is not surprising that the wet season is wetter the dry season, but it is notable that the wet season is drier than the 2014 dry season! Why is this information omitted?

Reply: Although the 2017 wet season showed slightly lower SWC values in the shallower soil layers in comparison to the 2014 dry season, the SWC values in the deeper layers were higher. We will add this information in the text.

[32] l.353: the API results don't tell the reader anything without proper explanation

Reply: Please see our reply to a previous comment.

[33] ll.359-360: two digits after comma reported for 18O – more than precision – should be avoided; add 'for' delta 18O, 'for' delta 2H

Reply: We will make these changes.

[34] l.382-384: because of the effect of clay material on extraction? (see comment before)

Reply: The soil water was isotopically distinct from rainfall due to mixing and soil evaporation processes.

[35] – same for ll. 387-388 l.417: the root biomass cannot be distinguished between species, right? (coffee vs. large trees?): : :that means that the created informative prior would be quite biased: : :.

Reply: Indeed, we were not able to distinguish between roots of coffee shrubs and shade trees, but we don't understand how this can have caused a bias in the prior information.

[36] ll.432-436: discussion Putting the rainfall amounts in the results section is debatable: : :it sure is something that was done during the study, but it is not directly related to the objectives. As Hydrologist, I personally would've liked to read these numbers earlier to put the words 'dry season', 'less than average' etc. in perspective.

Reply: We would like to refer the reviewer to Section 3.1., in which we provide the rainfall amounts for the dry and wet seasons sampled and compare these with the long-term data from 1970-2000.

[37] Discussion Ll.522-525: So in the wet season both trees and coffee use shallow water, because it's abundant. In the dry season, the trees use deep water – because they have deeper roots and water in deeper soil is easier accessible (low matric potential of soils). The coffee uses shallower water in the dry season. What is the reason? – the fact that coffee plants cannot grow deep roots? – or is it because they don't need so much water compared to the trees and don't need deep roots? – or, because the coffee plant has another strategy and its roots can extract water from drier soil compared to tree roots? or: : :.... This is not a criticism; this question is out of interest. I wonder then, if this is really 'complementary' water use as such?

Reply: Many of these issues have been addressed in the Section 4.1 in the Discussion, and yes, there are open questions that need further research as mentioned in the lines 596-597 and 601-603.

[38] ll.599- 600: Which recommendations based on their results would the authors give to coffee producers then? This would be a nice addition.

Reply: We would like to refer the reviewer to Section 4.3, in which we discuss the implications of our results and future research directions.

[39] ll.606-612: this is a bit contradictory, because in the presented example using this additionally information did not affect the results much (both uncertainty and general outcomes). So which variables should beincluded in the future? Are there others that

might be more suitable? Micronutrients? Soil moisture?....

Reply: As it is mentioned in the text, although our results did not change significantly by including or excluding the root and nutrient data (informative priors), exploring potential sources of water uptake using an informative and non-informative prior approach provided more confidence in our results. For other environments, the use of prior information may lead to different results and value to better understand water uptake patterns/processes (L604-610).

[40] Conclusions An experienced and well-known researcher a while ago gave me the advice: 'A good paper does not need a conclusion chapter – the reader draws them him/herself.' That stuck to me somehow. I think this is a good paper.

Reply: We believe that a conclusions section is essential for a paper, because it gives the reader a quick overview of the most important findings.

References

Araguás-Araguás L, Rozanski K, Gonfiantini R, Louvat D. 1995. Isotope effects accompanying vacuum extraction of soil water for stable isotope analyses. J. of Hydrology 168, 159–171.

Barbeta A, Jones SM, Clavé L, Wingate L, Gimeno TE, Fréjaville B, Wohl S, Ogée J. 2019. Unexplained hydrogen isotope offsets complicate the identification and quantification of tree water sources in a riparian forest. Hydrol. Earth Syst. Sci. 23, 2129–2146.

Muñoz-Villers LE, Holwerda F, Alvarado-Barrientos MS, Geissert D, Dawson TE. 2018. Reduced dry season transpiration is coupled with shallow soil water use in tropical montane forest trees. Oecologia 188, 303–317.

Ogle K, Tucker C, Cable JM. 2014. Beyond simple linear mixing models: Processâ ̆Řbased isotope partitioning of ecological processes. Ecological Applications 24, 181–195.

Orlowski N, Breuer L, McDonnell JJ. 2016. Critical issues with cryogenic extraction of soil water for stable isotope analysis. Ecohydrology 9, 1–5.

Schoonheydt RA, Johnston CT. 2015. Surface and interface chemistry of clay minerals. Developments in Clay Science 6, 139.

Vidal O, Dubacq B. 2009. Thermodynamic modelling of clay dehydration, stability and compositional evolution with temperature, pressure and $H_2O$ activity. Geochim. Cosmochim. Acta 73, 6544.

West AG, Patrickson SJ, Ehleringer JR. 2006. Water extraction times for plant and soil materials used in stable isotope analysis. Rapid Commun. Mass Sp. 20, 1317–1321.
* * *

---

## Author Comment (AC2) · 18 Oct 2019

We thank Adriá Barbeta for his positive and encouraging comments giving us the opportunity to further improve the article. Please find below our response to your comments.

[1] Although my general assessment of the manuscript is highly positive, I miss some caution regarding stable isotope techniques. While this is a well-established approach, recent studies pointed to methodological issues linked to fractionation processes within the soil matrix (Orlowski et al., 2018; Gaj et al., 2019; Oerter & Bowen, 2019; Oerter et al., 2019), along the soil-plant continuum (Vargas et al., 2017; Barbeta et al., 2019)

or within plant tissues (Zhao et al., 2016). Not all ecohydrological systems may be affected by those fractionation processes, and oxygen isotopes seem to still be highly reliable (Zhao et al., 2016; Vargas et al., 2017; Barbeta et al., 2019). Still, in Fig. 3, I observe that xylem water isotopes do not match very well with soil water isotopes from either depth. This is clearer for shade trees. A similar pattern arises in the deuterium excess boxplots. A thorough consideration of potential fractionation processes would require extensive additional analyses, which I think that it is not realistic to ask the authors to do. A more plausible solution is an explanation on why the authors think that fractionation processes are not relevant for their study. It might also be considered to run MixSIAR models separately for oxygen and hydrogen isotopes to check if there are significant discrepancies between them (as in Evaristo et al., 2017; Barbeta et al., 2019). As I said, it is known that fractionation processes do not affect in the same proportion oxygen and hydrogen isotopes. In any case, I believe that these emerging issues cannot longer be ignored by plant water source studies using stable isotopes.

Reply: We agree that fractionation process may, and can no longer be omitted/discussed in the types of data our study presents. In fact, one of our co-authors has been an advocate and champion of doing the best possible research to discover when such affects might play a role (see Brantley et al. 2017; Oshun et al. 2016; Penna et al. 2018). Calculating the isotopic composition range of xylem water and the considered sources across sampling periods and seasons, it is observed that all shade trees (-7.6 to -3.6 for $\delta$18O, and -65.5 to -32.2 for $\delta$2H) and coffee plants (-6.3 to -0.6 for $\delta$18O and -46.5 to -9.6 for $\delta$2H) fell within the range of the soil water pool (-11.1 to -0.9 for $\delta$18O, and -83.4 to -11.9 for $\delta$2H) during the 2014 dry season samplings (Fig. 3a). In the 2017 dry season samplings, we again observed a good isotopic match between the tree xylem water (-6.0 to -3.2 for $\delta$18O, and -56.7 to -34.5 for $\delta$2H) and the soil pore (-7.5 to -1.6 for $\delta$18O, and -54.8 to -19.0 for $\delta$2H). However, for the coffee plants, the xylem water (-4.4 to -1.1 for $\delta$18O and -39.6 to -7.9 for $\delta$2H) had more enriched $\delta$2H values in comparison to soil water (Fig. 3b). In the 2017 wet season sampling, a very small mismatch was detected in $\delta$2H between xylem water of coffee (-5.4 to -4.4

for $\delta18O$ and -42.2 to -34.5 for $\delta2H$) and soil water (-8.5 to -4.1 for $\delta18O$ and -70.5 to -37.5 for $\delta2H$), meanwhile the trees (-6.2 to -4.2 for $\delta18O$ and -60.6 to -45.6 for $\delta2H$) showed again a good overlap with soil water (Fig. 3c). Based on these results and following the reviewers suggestion, we will carry out some tests to evaluate the effects of deuterium fractionation, in particular for the coffee water samples, by running a simple mass balance approach using hydrogen isotopes only, although we are aware that single isotope ratio approach in multiple water source model could lead to erroneous results due to the overlap of feasible solutions, with poor constrained of uncertainties (see Parnell et al., 2010). In that case, Bayesian mixing models using both deuterium and oxygen isotopes could produce more reliable estimates and uncertainties (Evaristo et al. 2017). Finally, we will add additional explanations in the discussion about potential fractionation processes and their effects on the quantification of the plant water sources.

[2] Minor comments L38 It is not completely clear what does 'precipitation conditions' mean.

Reply: We will clarify this in the manuscript.

[3] L65 Species name (Cedrela odorata) should not be in capital letters.

Reply: Agree. We will make the correction.

[4] L191 The high clay content is likely to produce soil water isotopic fractionation (Oerter et al., 2014).

Reply: Since it has been shown that soil samples containing a high clay fraction might affect the quality of the soil water extraction, and therefore the isotopic composition of the bound water, several papers have suggested that investigations should now incorporate information of the soil hydro-physical properties, and more importantly for clayey soils, information about the cation exchange capacity (CEC), as Vidal and Dubacq (2009) have pointed out that the effect of this interlayered space/water in clay-rich soils

can be indirectly evaluated with CEC. For our study, we did determine other soil physical and chemical properties such as CEC. Therefore, we are going to incorporate this information in the manuscript to show that the contribution of this interlayer water bound in the clay mineral structure was small for our soils, and therefore of little significance for the entire isotopic composition of the extracted soil water, and for the mixing model results (see also our reply to a similar comment made by reviewer #1).

[5] L218 The sampling of different plant parts in coffee plants and shade trees (cores VS branches) could have led to a different proportion of internal plant water pools in the xylem water samples of each group.

Reply: Agree. However, for the coffee, it was not possible to collect a xylem core from the main stem as for the trees without considerable damage, because of the smaller stem diameter of the coffee plants. Therefore, to sample comparable plant xylem water pools between trees and coffee, segments (∼6 cm) of mature branches were cut near the main stem for the coffee plants.

[6] L223 I assume that bark was peeled off from coffee shrubs, too.

Reply: The bark from the branch segments of coffee shrubs was not peeled off (∼1mm in width around the segment), because doing so would have taken considerable time and thus potentially expose the sample to evaporation; we will add this information in the text.

[7] L298 Recent precipitation, especially in periods with relatively wet soil conditions, could in fact percolate faster towards deeper layers. So, rainfall is not necessarily representative of near surface soil water.

Reply: We totally agree and we have revisited this assumption in response to your comment and Reviewer #1 comments. Since each isotope sampling campaign was preceded by at least 6 days up to 22 days without or with minimum accumulated rainfall (< 5 mm) (L235-236), we acknowledge the difficulties to consider rainfall as a potential

source of near surface soil water. Therefore, we are going to use the isotopic composition of the soil at 5 cm depth as a source for near surface soil water. Thus, the discretization of the mixing model originally presented will be modified in the revised version. It is important to mention that we moved already to perform some statistical tests to define the new classification of the soil water pool. Based on the outcomes, the soil water pool will be divided in the following compartments: near surface water (5 cm depth), shallow (15 cm depth), intermediate (30 cm depth) and deep (average of 60-120 cm depth) soil water sources. Preliminary runs of the Bayesian mixing model using this new discretization and without or with the informative prior data continue to show a complementary water use strategy between trees and coffee plants during the dry and wet periods investigated.

[8] L304 The use of prior information is a very interesting point of the study.

Reply: We appreciate your comment.

References

Brantley SL, Eissenstat DM, Marshall JA, Godsey SE, Balogh-Brunstad Z, Karwan DL, Papuga SA, Roering, J, Dawson TE, Evaristo J, Chadwick O, McDonnell JJ, Weathers KC. 2017. Reviews and syntheses: On the roles trees play in building and plumbing the critical zone. Biogeosciences 14, 5115–5142.

Evaristo J, McDonnell JJ, Clemens J. 2017. Plant source water apportionment using stable isotopes: A comparison of simple linear, two‐compartment mixing model approaches. Hydrological Processes 31, 3750–3758.

Oshun J, Dietrich WE, Dawson TE, Fung I. 2016. Dynamic, structured heterogeneity of water isotopes inside hillslopes. Water Resources Research 52, 164–189.

Parnell AC, Inger R, Bearhop S, Jackson AL. 2010. Source partitioning using stable isotopes: Coping with too much variation. PloS One 5, e9672.

Penna D, Hopp L, Scandellari F, Allen ST, Benettin P, Beyer M, Geris J, Klaus J, Mar-

shall JD, Schwendenmann L, Volkmann THM, von Freyberg J, Amin A, Ceperley N, Engel M, Frentress J, Giambastiani Y, McDonnell JJ, Zuecco G, Llorens P, Siegwolf RTW, Dawson TE, Kirchner JW. 2018. Ideas and perspectives: Tracing terrestrial ecosystem water fluxes using hydrogen and oxygen stable isotopes – challenges and opportunities from an interdisciplinary perspective. Biogeosciences 15, 6399–6415.

Vidal O, Dubacq B. 2009. Thermodynamic modelling of clay dehydration, stability and compositional evolution with temperature, pressure and H2O activity. Geochim. Cosmochim. Acta 73, 6544.

---

## Author Comment (AC3) · 18 Oct 2019

We thank Daniele Penna for his positive comments on our manuscript. Please find below our response to each comment.

Minor comments and technical corrections

[1] 42-43. This sentence is not immediate to understand without reading the paper. I suggest rephrasing.

Reply: The sentence will be rephrased.

[2] 45. Double negation (limitation: : :absent): I suggest simplifying the sentence.

[Figure]

Reply: We will correct this.

[3] 64. Complex sentence, rephrase.

Reply: The sentence will be rephrased.

[4] 149. I suggest to change into ": : :prevails over competition: : :" or, in any case, to include both the terms "complementary" and "competition" because the latter is logically linked to the second research question.

Reply: We will follow the suggestion.

[5] 268. Did you consider using the Normalized Antecedent Precipitation Index (NAPI, Heggen, 2001), instead of API? Heggen, R.J., 2001. Normalized antecedent precipitation index. J. Hydrol. Eng.

Reply: No, we did not consider it but we can always explore the method. Thank you for the suggestion.

[6] 385-405. I suggest to condense this part and let the figures talk for themselves.

Reply: Ok. We will follow your recommendation.

[7] Fig. 3. Caption: why panel (c) shows the GMWL whereas panels (a) and (b) the LMWL?

Reply: We will correct the text in the figure caption to say that in panels (a), (b) and (c), the solid line represents the GMWL.

[8] Fig. 4. I suggest to replace "(a)" and "(b)" with "2014" and "2017" for more immediate understanding.

Reply: We will make this replacement.

[9] Fig. 5. What do error bars represent? Why are there only in panel (a) and not in panel (b)?

Reply: The bars represent the standard deviation; we will add this information in the figure caption. These bars are not showed in the panel (b) because the values in the y axis were normalized and expressed as ratio to their maximum values.

[10] Fig. 8. I think that the result and discussion build around this figure should be taken with a bit of caution because based on few point only. I suggest to discuss this limitation in the manuscript.

Reply: Agree. We are going to be more careful with this result and mention its limitation in the discussion.

---

## Author Response (AR1)

Matthias Beyer (Referee)
matthias.beyer@bgr.de

Thank you for letting me review the manuscript hess-2019-329 'Coffee and shade trees show complementary use of soil water in a traditional agroforestry ecosystem' by Muñoz-Villers et al. I enjoyed reading. In their work, the authors investigate water uptake depths of large shade trees and coffee trees during two dry seasons and one wet season using water stable isotopes and a Bayesian mixing model. They find that coffee and shade trees show complementary water use patterns, i.e. preferential water use by coffee and deep-water use by the shade trees. During the wet season, both groups shift to shallower resources. Without doubt, this manuscript is well-prepared and written. The structure is clear, research questions are stated concisely, and the Introduction provides a thorough overview on the topic. The graphics are suitable. I like the study and the topic is interesting. Also – and this is the main scientific contribution of the paper – it is great to see that the authors integrated priors and used root information and macronutrient distributions for that. However, apart from that, the novelty and innovation of the study is limited. I also have a couple of rather major concerns about the methods used in the paper related to the soil water extraction and mixing model. The former might be answered, but the latter might require some more effort. I elaborate on those below. Minor comments are summarized further down. Good luck and all the best, Matthias Beyer

**We thank Matthias Beyer for his positive and constructive comments which allow us to further improve the article. Please find below our response to each of the comments.**

**Major points:** l.249-250: was complete extraction somehow validated? Also note that clay-rich soils need higher extraction temperatures (see recent (Gaj et al., 2017; Orlowski et al., 2016) papers on mineral mediated isotope fractionation). Using a water bath at 100_C might result in an offset in isotope compositions and lead to errors/ uncertainty in the mixing model (the reservoir of water that is extracted would not equal the reservoir that is available to plants). The authors state at one point that there was an offset of the values towards more depleted – this is exactly what would happen and was observed in other studies when clay was an issue. This issue should be at least discussed.

**Reply:** Validation of complete extraction. We did not check whether all water was extracted using a gravimetric water content assessment. However, according to the findings of Araguas-Araguas et al. (1995) and West et al. (2006), extractions do not have to reach full completion (i.e., all water extracted) to obtain an unfractionated and, therefore, isotopically consistent value. Experiments have shown that the isotope value of any extracted water increased quickly during the first 20-75 minutes of extraction, after which the isotope value of the extracted water remained constant regardless of further increases in extraction time.

The time at which this threshold is reached is the minimum extraction time (Tmin) required to obtain an isotopically unfractionated water sample, and once Tmin is reached, only a very small amount (microliters) of water may remain in the sample. Recently, Orlowski et al. (2013) showed that even if the extraction is conducted until what they claimed was complete, the isotopic signature may not be recovered from different soil types.

The Tmin value varies with the source material. West et al. (2016) showed that woody stems required the longest extraction times (60–75 min), while values of Tmin were shorter for soil (40 and 30 min for clay and sand soil textures, respectively). Following West et al. (2016), we used the same extraction time for stems and soils (60-70 min)(Section 2.4, L250).

Clay-rich soils need higher extraction temperatures. Apart from extraction duration, the literature has shown that the extraction temperature "might" have an impact in the soil isotopic composition. Araguas-Araguas et al. (1995) showed that a highly mobile water reservoir that is weakly bound to soil particles can exist (especially in clay-rich soils where interlayered water can be present), and remains largely intact at extraction temperatures < 100°C. More recently, the studies of Orlowski et al. (2016) and Schoonheydt and Johston (2015) have discussed whether the extraction temperature should be increased. However, there has been no systematic investigation that clearly identified the driving forces that might cause an isotope effect on the isotopic composition of the extracted soil water. Since it has been shown that soil samples containing a high clay fraction might affect the quality of the soil water extraction, and therefore the isotopic composition of the bound water, several papers have suggested that investigations should now incorporate information of the soil hydro-physical properties, and more importantly for clayey soils, information about the cation exchange capacity (CEC), as Vidal and Dubacq (2009) have pointed out that the effect of this interlayered space/water in clay-rich soils can be indirectly evaluated with CEC. For our study, we did determine other soil physical and chemical properties such as CEC. We have incorporated this information in the revised manuscript now (Section 2.5, L278-283) to show that the contribution of this interlayer water bound in the clay mineral structure was small for our soils (Section 3.4, L446-447; Table 4), and therefore of little significance for the entire isotopic composition of the extracted soil water (Section 4.1, L519-535).

Importantly, we did state that the values of $\delta 2H$ and $\delta 18O$ in plant xylem water ($-40.8 \pm 15.0$‰ and $-4.6 \pm 1.6$‰, respectively) were on average more positive in comparison to bulk soil water ($-46.7 \pm 16.4$‰ and $-6.0 \pm 2.3$‰, respectively) (L386-388 in the original ms); however, comparing the isotopic composition range of xylem water and the soil water sources across sampling periods, we observed a good isotopic match between the tree xylem water and the soil water, while for the coffee plants, the xylem water had more enriched $\delta 2H$ values in comparison to soil water. We have added this information in the revised version (Section 3.3, L425-433). To evaluate the effects of deuterium fractionation on coffee water sources, we compared the relative contribution of each water source obtained via the single isotope ($\delta 2H$) mixing model with those obtained via the informative prior distribution model. The results of these tests have been presented in Section 3.6. Finally, we have discussed this issue and its potential effect on the quantification of the plant water sources (Section 4.1, 519-558).

Another question (but this is more general) related to the cryogenic extraction is why such long extraction times are needed (I know, West et al. 2006 propose that). I think one part of that is related to the relatively low extraction temperature, but still. The extractable water should be leaving the sample side very fast given the low volumes (even under 100 _C) – waiting longer would not evaporate more water from the sample side unless the temperature is increased further.

**Reply:** Please see our reply to your previous comment.

l. 297-300: These assumptions need to be validated/proven. Why was not the soil water isotope composition of the first 5 cm used directly? I guess in order to account for water that was taken up by the plant before the actual sampling date?

**Reply:** We have revisited this assumption. Since each isotope sampling campaign was preceded by at least 6 days up to 22 days without or with minimum accumulated rainfall (< 5 mm), we acknowledge the difficulties with this approach. Hence, following the reviewer' suggestion, we decided to take the isotopic composition of the soil water at 5 cm depth as representative of near surface soil water. As a result, the discretization of the mixing model originally presented has changed in the revised version; Methods (Section 2.7, L303-315) and Results (Sections 3.2, 3.3 and 3.5; Table 2 and 3; Figure 3, 6 and 7) have changed accordingly. Please also see our reply to the next comment.

How was the classification used for the mixing model decided? Slightly above and below the zero-flux plane, the isotope composition of soils normally changes drastically during dry periods: : :for clay this is often in the first 15 cm soil depth. The 30 -120 cm depth were isotopically similar? In my understanding, the discretization used in the mixing model should be done after the isotope depth profiles are evaluated and backed up by statistical measures of differences between different depths. After checking the supplementary data, I'm really doubting the discretization used. There are partially huge differences of the isotope values of the soil profiles between 30 and 120 cm. And how about 15-30 cm? – was the isotope information of this depth not used at all? (in that case, the mixing model is missing a source which violates the mixing model requirements). I refer, once again, to the Rothfuss et al. publication, which might help to address these issues.

**Reply:** The classification used in the mixing model was based on the changes in the isotopic composition of soil water and the changes in the root and nutrient distributions along the soil profile. In the original manuscript, we divided the soil water pool in two compartments: shallow (5-15 cm depth) and deep soil (30-120 cm depth) sources. In each campaign, we sampled the soils for isotopes at the following depths: 5, 15, 30, 60, 90 and 120

cm. Further, we classified the soil isotope data collected at 5 and 15 cm as shallow and those obtained at 30, 60, 90 and 120 cm depth as deep. Thus, the potential tree water sources that we considered were restricted to these categories and data. There are other examples in the literature in which the evaluation of the relative contribution of soil water sources to plant uptake has been restricted to particular groups of soil depth (cf. Barbeta et al. 2019), without violating the mixing model requirements.

However, since the isotopic composition at 5 cm depth was used as the near surface water source following the reviewers' suggestion, we ran again the statistical tests to define the new classification of the soil water pool. Based on the results of these tests, the soil water pool was divided in the following compartments: near surface (5 cm depth), shallow (15 cm depth), intermediate (30 cm depth) and deep (average of 60-120 cm depth) soil water sources (Section 2.7, L303-308).

**Minor points:** - Since many different analysis were carried out with the soil and plant samples, this could be summarized in a table nicely. - It would have been easy and interesting to check the uptake depths of the large trees separately and not lumping them. (but maybe not of interest for the study)
**Reply:** Since these analyses are already described in detail in the text, we consider it redundant to add a table. With regard to the uptake depth, we were unable to distinguish between roots of coffee shrubs and shade trees, as well as between the roots of the different species of shade trees. We have now added this information to the text (Section 2.6, L293-294).

- I suggest strong discussion of the use of informative priors and putting a more general focus on this aspect, as this is the key scientific/methodological novelty in this paper in my opinion.
**Reply:** We have improved this in the discussion to stress the importance of using informative priors in the mixing models (Section 4.1, L509-518).

- (more a comment): It would have been interesting to have water potential measurements in both soils and trees, because those could really constrain the possible uptake depths.
**Reply:** Yes, we agree that such data would have been interesting. In a follow-up study, we have been doing water potential measurements at the time of sample collection for isotope analysis.

Abstract l.27: Providing the rainfall amounts in addition to the year would be nice; in addition, it would be nice if the authors could state the type of environment of the study (e.g. semi-arid, tropical,: : :)
**Reply:** We have added this information in the Abstract (L23).

ll.35/36: the percentages are the mean? median? I suggest adding a +/- xx % notation accounting for uncertainty
**Reply:** The percentages are mean values; we have added the +/- % standard deviation (L35-37).

l.39: short-term wetness status? Do the authors mean that the uptake depth is not influenced by small rain events? This sentence is not easy to understand, I suggest rephrasing
**Reply:** The sentence was rephrased for clarification (L37-38).

ll.39-41: this sentence needs to be rephrased. The terms near surface vs. much shallower are confusing the reader (5 and 15 cm are both shallow). Perhaps 'upper five centimeter'?
**Reply:** We used the terms mentioned above (i.e., near surface for 5 cm depth and shallow for 15 cm depth).

ll.42-43: the spatial segregation mentioned, is it due to the different rooting depths of the studied plants? Was this validated somehow?
**Reply:** Please see our reply to a similar previous comment.

l.44: plant-soil water uptake? Confusing phrase. Do the authors mean 'root water uptake patterns/depths'? I feel like a concluding sentence is missing in the abstract. What are the implications of the study? What novel things were found out? Is 120 cm the max. rooting depth??? Uptake depth vs. rooting depth? (coffee shallow, others deep)

**Reply:** Yes, we mean root water uptake patterns. We have changed this (L42). Also, we rephrased this sentence to represent our main conclusion (L41-43). The implications of our study are presented in Section 4.4 (Implications and future direction) in the Discussion. The contribution (novelty) of this research has been argued in the Introduction and the Discussion sections. 120 cm was the deepest potential water source that we examined. It is unclear what the reviewer means with the question about water uptake vs. rooting depth with regard to line 44 (line 42 in the revised ms).

Introduction I really like the way the introduction is written (clear and concise). The Bayesian mixing model needs to be addressed though. The word is only mentioned once, and some readers might not know what it even is. At the end of the introduction, sentence is missing highlighting the importance and novelty of this research.
**Reply:** We have provided more background information about Bayesian mixing models and highlight the novelty of including priors for the quantification of plant water sources (Section 1, L88-93).

l.55 and l.73: 'soil resources' sounds odd: : :can the authors specify please?
**Reply:** We have been more clear.

l.87: However,
**Reply:** The suggestion has been followed.

l. 90-92: please note that mixing models are also frequently criticized, (Rothfuss and Javaux, 2016)
**Reply:** We are aware that mixing models have been criticized; however, they have several advantages over other methods. That is, they allow for determining the likelihood of the different water sources available to plants using a robust statistical approach and they allow for the incorporation of biophysical parameters (e.g., root and nutrient data) as informative priors (Muñoz-Villers et al. 2018).

l.92: 'Although rarely implemented' – do the authors have examples where it was implemented? (this is out of interest)
**Reply:** To our knowledge, Muñoz-Villers et al. (2018) have been the only ones to use nutrient and root distribution data as priors to better inform a Bayesian mixing model. We have added this reference to the text (Section 1, L95).

l.143: micrometeorological measurements (which)
**Reply:** We have changed this to "microclimatic measurements" (L143). The list of the microclimatic variables that were measured are provided in Section 2.2.

l. 146: nice the authors are implementing priors. See related publication where this was suggested (and also MixSIAR was used): (Beyer et al., 2018). You don't have to cite us but maybe it helps for some explanation in the authors manuscript.
**Reply:** Thank you for the recommendation. We have included this reference in our Introduction (L91).

l.151/152: The answer to question no. 2 is not reflected in the abstract Materials/Methods
**Reply:** We present the results of the two dry seasons investigated (the near normal and the more pronounced one), in the abstract (L26-27; L33), therefore we did answer the question #2. With regard to the Materials/Methods, in Section 2.3 we mentioned that the dry season of 2017 was warmer and drier offering the opportunity to examine the vegetation responses under more pronounced dry conditions.

l. 168: on an; is there no data after 2000 for rainfall? This seems like it's likely to have changed meanwhile
**Reply:** Indeed, there are no data after 2000. And we don't have the data ourselves to determine if there have been any changes in rainfall.

l.214: 'carried out' rather than 'performed'?
**Reply:** The change has been made.

l.218-222: how many replicates per individual were taken? (same later for coffee and the soil samples)
Reply: This information is given in Tables 1, 2 and 3. For the coffee, the number of replicates is also provided in the text (L224-228). For the trees, soil and rain samples, we have added this information in the text (L218-219; L232; L240).

l.232-233: 'Auger sampling points were located so that each of the sampled shade trees and coffee plants had a total of three soil sampling points within their 3 m radius.' – If it was sampled at only three different locations (see sentence before), so it means that all the trees had the three sampling points in their 3m radius? That seems odd. Can the authors please check if this phrasing is correct here?
Reply: We have rephrased the sentence for clarification (L232-233).

l.247: refrigerated – was any mold developing on the samples? This can affect isotope ratios
Reply: Some mold had developed on some of the samples of the trees and coffee shrubs, but this does not affect the xylem isotope ratios.

l.268/269: What is API – if it is not a common method, it needs to be explained briefly.
Reply: API stands for antecedent precipitation index and it was calculated following the method of Viessman et al. (1989) (L267-269). It is actually a common hydrological metric used to quantify the antecedent precipitation conditions (7 or 15 days) prior to a rainfall event, sampling date, etc.

l.304/305: It would be very appreciable to the community I believe if the authors explain how the priors were determined and implemented into MixSIAR as this is not something that has been done often.
Reply: We have now added this information in the Supplementary Material.

Results
l.321: I see a point in putting this as result, but this is nothing that belongs to the objectives of the study as such. I suggest including it into the methods chapter. In many hydrologic and soil studies variables such as rainfall and soil moisture are the basis and not highlighted as results.
Reply: This section characterizes the hydrometeorogical conditions during the two dry seasons (2014 and 2017) and the wet season (2017) studied. Since one of our objectives was to determine the sources of plant water under different soil water availability conditions, we consider it important to present this information as part of the Results section.

l. 335: Definition of normal vs. below-average dry season: In fact, both dry seasons sampled were below average, 2014 was about 20% lower (323 mm vs. 389 mm normal) and the 2016/17 one 40%....not sure if I would consider 20% below average a 'normal' year.
Reply: Indeed, rainfall during the 2013-2014 dry season was about 20% lower than normal. Hence, following the suggestion of the reviewer we refer to this season as "near normal" in the revised manuscript.

l.351-353: it is not surprising that the wet season is wetter the dry season, but it is notable that the wet season is drier than the 2014 dry season! Why is this information omitted?
Reply: Although the 2017 wet season showed slightly lower SWC values in the shallower soil layers in comparison to the 2014 dry season, the SWC values in the deeper layers were higher. We have added this information in the text (L366-368).

l.353: the API results don't tell the reader anything without proper explanation
Reply: Please see our reply to a previous comment.

ll.359-360: two digits after comma reported for 18O – more than precision – should be avoided; add 'for' delta 18O, 'for' delta 2H
Reply: We have made the changes.

l.382-384: because of the effect of clay material on extraction? (see comment before)
Reply: The soil water was isotopically distinct from rainfall due to mixing and soil evaporation processes. Please also see our reply to one of your previous related questions.

– same for ll. 387-388 l.417: the root biomass cannot be distinguished between species, right? (coffee vs. large trees?): : :that means that the created informative prior would be quite biased: : :.
Reply: Indeed, we were not able to distinguish between roots of coffee shrubs and shade trees. As we mentioned earlier, we have included this information in the text (L293-294). However, we do not understand how this can have caused a bias in the prior information.

ll.432-436: discussion Putting the rainfall amounts in the results section is debatable: : :it sure is something that was done during the study, but it is not directly related to the objectives. As Hydrologist, I personally would've liked to read these numbers earlier to put the words 'dry season', 'less than average' etc. in perspective.
Reply: We would like to refer the reviewer to Section 3.1., in which we provide the rainfall amounts for the dry and wet seasons sampled and compare these with long-term data from 1970-2000.

Discussion Ll.522-525: So in the wet season both trees and coffee use shallow water, because it's abundant. In the dry season, the trees use deep water – because they have deeper roots and water in deeper soil is easier accessible (low matric potential of soils). The coffee uses shallower water in the dry season. What is the reason? – the fact that coffee plants cannot grow deep roots? – or is it because they don't need so much water compared to the trees and don't need deep roots? – or, because the coffee plant has another strategy and its roots can extract water from drier soil compared to tree roots? or: : :.... This is not a criticism; this question is out of interest. I wonder then, if this is really 'complementary' water use as such?
Reply: Many of these issues have been addressed in Section 4.2 in the Discussion, and yes, based on our findings, shade trees and coffee plants are complementary in their use of soil water.

ll.599- 600: Which recommendations based on their results would the authors give to coffee producers then? This would be a nice addition.
Reply: We would like to refer the reviewer to Section 4.4, in which we discuss the implications of our results and future research directions.

ll.606-612: this is a bit contradictory, because in the presented example using this additionally information did not affect the results much (both uncertainty and general outcomes). So which variables should beincluded in the future? Are there others that might be more suitable? Micronutrients? Soil moisture?....
Reply: As it is mentioned in the text, although our results did not change significantly by including or excluding the root and nutrient data (informative priors), exploring potential sources of water uptake using an informative and non-informative prior approach provided more confidence in our results. For other environments, the use of prior information may lead to different results and value to better understand processes that lead to differences in the depth of plant water uptake (Section 4.1, L509-518).

Conclusions An experienced and well-known researcher a while ago gave me the advice: 'A good paper does not need a conclusion chapter – the reader draws them him/herself.' That stuck to me somehow. I think this is a good paper.
Reply: We believe that a conclusions section is essential for a paper, because it gives the reader a quick overview of the most important findings.

Review of manuscript HESS-2019-329

General comment
This study analyses plant water source partitioning in a coffee agroforestry system along seasons with contrasting soil moisture conditions. For that, the authors applied stable isotope techniques and Bayesian mixing models (MixSIAR) in order to test for the complementary use of soil water in space and time by coffee plants and shade trees. The importance of the study comes from the fact that ecohydrological relations in this type of traditional agroforestry systems are completely unknown, in contrast to those of intensive monospecific plantations. A novel aspect of the study is the inclusion of root and nutrient distributions within the framework of stable isotope mixing models, which is a usually underestimated capability of such models. That should improve their accuracy since plant water source partitioning is obviously constrained by root distribution and soil profiles of nutrient availability. Overall, this is a welldesigned, rigorous study, that is also clearly presented and well-written. Methods and results are concisely described and figures and tables are easy to interpret. Similar studies of plant water source partitioning are numerous, so it could be said that this study is not especially original. However, I find valuable to report this type of data from regions where they are scarce (i.e. Central and South America or Africa, see Barbeta & Peñuelas, 2017; Evaristo & Mcdonnell, 2017).

**We thank Adriá Barbeta for his positive and encouraging comments giving us the opportunity to further improve the article. Please find below our response to your comments.**

Although my general assessment of the manuscript is highly positive, I miss some caution regarding stable isotope techniques. While this is a well-established approach, recent studies pointed to methodological issues linked to fractionation processes within the soil matrix (Orlowski et al., 2018; Gaj et al., 2019; Oerter & Bowen, 2019; Oerter et al., 2019), along the soil-plant continuum (Vargas et al., 2017; Barbeta et al., 2019) or within plant tissues (Zhao et al., 2016). Not all ecohydrological systems may be affected by those fractionation processes, and oxygen isotopes seem to still be highly reliable (Zhao et al., 2016; Vargas et al., 2017; Barbeta et al., 2019). Still, in Fig. 3, I observe that xylem water isotopes do not match very well with soil water isotopes from either depth. This is clearer for shade trees. A similar pattern arises in the deuterium excess boxplots. A thorough consideration of potential fractionation processes would require extensive additional analyses, which I think that it is not realistic to ask the authors to do. A more plausible solution is an explanation on why the authors think that fractionation processes are not relevant for their study. It might also be considered to run MixSIAR models separately for oxygen and hydrogen isotopes to check if there are significant discrepancies between them (as in Evaristo et al., 2017; Barbeta et al., 2019). As I said, it is known that fractionation processes do not affect in the same proportion oxygen and hydrogen isotopes. In any case, I believe that these emerging issues cannot longer be ignored by plant water source studies using stable isotopes.
**Reply:** We agree that fractionation processes may, and can no longer be omitted/discussed in the types of data our study presents. In fact, one of our co-authors has been an advocate and champion of doing the best possible research to discover when such affects might play a role (see Brantley et al. 2017; Oshun et al. 2016; Penna et al. 2018).
Calculating the isotopic composition range of xylem water and the considered sources across sampling periods and seasons, it is observed that all shade trees (-7.6 to -3.6 for $\delta18O$, and -65.5 to -32.2 for $\delta2H$) and coffee plants (-6.3 to -0.6 for $\delta18O$ and -46.5 to -9.6 for $\delta2H$) fell within the range of the soil water pool (-11.1 to -0.9 for $\delta18O$, and -83.4 to -11.9 for $\delta2H$) during the 2014 dry season samplings (Fig. 3a)
In the 2017 dry season samplings, we again observed a good isotopic match between the tree xylem water (-6.0 to -3.2 for $\delta18O$, and -56.7 to -34.5 for $\delta2H$) and the soil pore (-7.5 to -1.6 for $\delta18O$, and -54.8 to -19.0 for $\delta2H$). However, for the coffee plants, the xylem water (-4.4 to -1.1 for $\delta18O$ and -39.6 to -7.9 for $\delta2H$) had more enriched $\delta2H$ values in comparison to soil water (Fig. 3b).
In the 2017 wet season sampling, a very small mismatch was detected in $\delta2H$ between xylem water of coffee (-5.4 to -4.4 for $\delta18O$ and -42.2 to -34.5 for $\delta2H$) and soil water (-8.5 to -4.1 for $\delta18O$ and -70.5 to -37.5 for $\delta2H$), meanwhile the trees (-6.2 to -4.2 for $\delta18O$ and -60.6 to -45.6 for $\delta2H$) showed again a good overlap with soil water (Fig. 3c). We have added this information in the Results (Section 3.3, L423-431), and based on these results, we carried out some tests to specifically evaluate the effects of deuterium fractionation on coffee water sources by running a simple mass balance approach using hydrogen isotope ratios only in the MixSIAR model. The results of these tests have been presented in Section 3.6.

Finally, we have discussed this issue and its potential effect on the quantification of the plant water sources (Section 4.1, 519-558).

Minor comments

L38 It is not completely clear what does 'precipitation conditions' mean.

**Reply:** We have rephrased the sentence for clarification (Abstract; L37-38).

L65 Species name (Cedrela odorata) should not be in capital letters.

**Reply:** Agree. We have made the correction (L66).

L191 The high clay content is likely to produce soil water isotopic fractionation (Oerter et al., 2014).

**Reply:** Since it has been shown that soil samples containing a high clay fraction might affect the quality of the soil water extraction, and therefore the isotopic composition of the bound water, several papers have suggested that investigations should now incorporate information about the soil hydro-physical properties. For clayey soils, information about the cation exchange capacity (CEC) should be given, as Vidal and Dubacq (2009) have pointed out that the effect of this interlayered space/water in clay-rich soils can be indirectly evaluated with CEC. For our study, we did determine other soil physical and chemical properties such as CEC. Therefore, we have incorporated this information in the revised manuscript (Section 2.5 in the Methods: L278-283; Section 3.4 in the Results: L446-447 and Table 4) to show that the contribution of this interlayer water bound in the clay mineral structure was small for our soils, and therefore of little significance for the entire isotopic composition of the extracted soil water (Section 4.1 in the Discussion: L519-535). See also our reply to a similar comment made by reviewer #1.

L218 The sampling of different plant parts in coffee plants and shade trees (cores VS branches) could have led to a different proportion of internal plant water pools in the xylem water samples of each group.

**Reply:** Agree. However, due to their much smaller size for the coffee, it was not possible to collect a xylem core from the main stem of the coffee plants without inflicting major damage. Therefore, to sample comparable plant xylem water pools between trees and coffee, segments (~6 cm) of mature branches were cut near the main stem of the coffee plants.

L223 I assume that bark was peeled off from coffee shrubs, too.

**Reply:** The bark (~ 1mm thick) from the branch segments of the coffee shrubs was not peeled off, because doing so would have taken considerable time and thus potentially expose the sample to evaporation; we have included this information in the Methods (L222-224) and also their potential effects on the enrichment observed in the deuterium composition of the coffee xylem water (Section 4.1 in the Discussion: L536-558).

L298 Recent precipitation, especially in periods with relatively wet soil conditions, could in fact percolate faster towards deeper layers. So, rainfall is not necessarily representative of near surface soil water.

**Reply:** We agree and we have revisited this assumption in response to your comment and a similar comment of Reviewer #1. Since each isotope sampling campaign was preceded by at least 6 days up to 22 days without or with minimum accumulated rainfall (< 5 mm), we acknowledge the difficulties with this approach. Therefore, we decided to take the isotopic composition of the soil water at 5 cm depth as representative of near surface soil water. As a result, the discretization of the mixing model originally presented has changed in the revised version; Methods (Section 2.7, L303-315) and Results (Sections 3.2, 3.3 and 3.5; Table 2 and 3; Figure 3, 6 and 7) have changed accordingly.

L304 The use of prior information is a very interesting point of the study.

**Reply:** We appreciate your comment.

Daniele Penna (Referee)
daniele.penna@unifi.it

Review of the manuscript 'Coffee and shade trees show complementary use of soil water in a traditional agroforestry ecosystem' by Muñoz-Villers et al. (hess-2019-329)

General comment
Reading this manuscript was pleasant. I particularly liked the idea of including the distribution of roots and nutrients in the mixing model approach, and I think that this point should be better stressed in the manuscript.

Overall, I fully agree with the comments provided by the reviews of Matthias Beyer and Adrià Barbeta, and I have only some minor comments to add.
Good job!
Daniele

**We thank Daniele Penna for his positive comments on our manuscript. Please find below our response to each comment.**

Minor comments and technical corrections

42-43. This sentence is not immediate to understand without reading the paper. I suggest rephrasing.
**Reply:** We have rephrased the sentence for clarification (L42-43)

45. Double negation (limitation: : :absent): I suggest simplifying the sentence.
**Reply:** We have corrected the sentence (L44-45).

64. Complex sentence, rephrase.
**Reply:** We have rephrased the sentence (L64-67).

149. I suggest to change into ": : :prevails over competition: : :" or, in any case, to include both the terms "complementary" and "competition" because the latter is logically linked to the second research question.
**Reply:** We have followed the suggestion (L148-149).

268. Did you consider using the Normalized Antecedent Precipitation Index (NAPI, Heggen, 2001), instead of API? Heggen, R.J., 2001. Normalized antecedent precipitation index. J. Hydrol. Eng.
**Reply:** We did not consider it but reading about this method we learned that NAPI utilizes the same antecedent daily precipitation record as does API in determining antecedent moisture conditions. The difference is that NAPI treats precipitation earlier in the day of a specific event (i.e. storm runoff or in our case the isotope sampling) as antecedent to the event. Since our samplings were conducted in dry days, the use of NAPI would not make a difference.

385-405. I suggest to condense this part and let the figures talk for themselves.
**Reply:** We have followed the suggestion. This part has been reduced in the revised version (L400-414).

Fig. 3. Caption: why panel (c) shows the GMWL whereas panels (a) and (b) the LMWL?
**Reply:** We have corrected the caption of Figure 3.

Fig. 4. I suggest to replace "(a)" and "(b)" with "2014" and "2017" for more immediate understanding.
**Reply:** We have made the change.

Fig. 5. What do error bars represent? Why are there only in panel (a) and not in panel (b)?
**Reply:** The bars represent the standard deviation; we have added this information in the figure caption. These bars are not showed in the panel (b) because the values in the *y* axis were normalized and expressed as ratio to their maximum values.

Fig. 8. I think that the result and discussion build around this figure should be taken with a bit of caution because based on few point only. I suggest to discuss this limitation in the manuscript.
**Reply:** Agree. We have mentioned this limitation in the discussion (L634).

Schoonheydt RA, Johnston CT. 2015. Surface and interface chemistry of clay minerals. Developments in Clay Science 6, 139.

[revised manuscript text omitted]

**Construction of the informative prior in the MixSIAR model: parameters and information**

Macronutrients (N, P, K) and root biomass data collected at the different depths were first grouped (averaged) in order to match the soil depth groups that represent the four potential plant water sources: Near Surface (5 cm), Shallow (> 5 to 20 cm), Intermediate (> 20 to 40 cm) and Deep (> 40 to 120 cm). Importantly, no significant difference was found between the dry and wet season nutrient profiles for 2017. Next, the nutrient and root biomass distributions were normalized such that the sum of the values for all the depths was 100. Then, for each sampling year, a composite depth distribution was constructed by averaging the N, P, K and root biomass profiles. The composite distribution was the prior probability for each source used in the Bayesian mixing model ("informative" approach). The resulting prior proportions used for the 2014 sampling dates were: Near Surface = 58, Shallow Soil = 15, Intermediate = 17 and Deep Soil = 10. For 2017 samplings, the following proportions were used: Near Surface = 53, Shallow Soil = 14, Intermediate = 17 and Deep Soil = 16. This configuration produced sharp proportions for each source, contrasting with those prescribed in the "non-informative" prior distribution (i.e. all the combinations of proportions of water sources were equally likely) as observed in Figure S3 for the 2014 dry season simulations.

[Figure]

**Figure S3.** Example of the prior distribution probability (informative approach) vs. the non-informative distribution used for the 2014 dry season samplings.

**Table S1.** Relative contributions of the different water sources to plant xylem water (mean ± SD) per species and for the three sampling dates performed in 2014 dry season. Contributions were derived with the MixSIAR Bayesian mixing model framework, using the 'informative' prior approach

[revised manuscript text omitted]

The water source that contributes more to tree transpiration is highlighted in bold for each species and sampling date.

**Table S7.** Relative contributions of the different water sources to coffee xylem water (mean ± SD) for the samplings performed in 2014 and 2017 dry seasons and 2017 wet season. Contributions were derived with the MixSIAR using the single-isotope ($\delta^2$H) mixing model

| | *C. arabica* | | | | | | |
|---|---|---|---|---|---|---|---|
| | 2014 | | | 2017 | | | |
| | Sampling 1 (Jan. 23) | Sampling 2 (Apr. 11) | Sampling 3 (Apr. 26) | Sampling 1 (Feb. 27) | Sampling 2 (Apr. 5) | Sampling 3 (May. 20) | Sampling 4 (Aug.4) |
| Near surface water | 0.20±0.29 | 0.41±0.44 | **0.75±0.36** | **0.92±0.21** | **0.99±0.04** | **0.94±0.20** | **0.98±0.04** |
| Shallow soil water | **0.73±0.38** | **0.54±0.46** | 0.22±0.37 | 0.05±0.18 | 0.01±0.04 | 0.05±0.20 | 0.01±0.03 |
| Intermediate soil water | 0.04±0.09 | 0.03±0.07 | 0.02±0.05 | 0.02±0.10 | 0.00±0.01 | 0.01±0.02 | 0.00±0.02 |
| Deep soil water | 0.04±0.08 | 0.02±0.05 | 0.02±0.03 | 0.01±0.03 | 0.00±0.01 | 0.01±0.02 | 0.01±0.03 |

The water source that contributes more to tree transpiration is highlighted in bold for each sampling date.

---

## Author Response (AR2)

Please find below our reply to the corrections:

1.) Should it be "two investigated dry seasons"?
Reply: The change has been made (L33)

2.) Line 313: Is this a typo or why were the grouping of isotope ratio (Line 305) and macronutrients and roots done using different depth groups? How can this be then used in the same mixing analysis?
Reply: Yes, this is a typo. Thank you for the observation. The correction has been made (L313-314)

.) Line 469: Should it probably be "sampling dates"?
Reply: Correct. The change has been made (L469).

.) Line 522: references are not correct formatted
Reply: The changes have been made (L522).

.) 621: Change to "On the other hand"
Reply: The change has been made (L621)

.) Line 649: Is there a ", respectively" missing at the end of this sentence? If not, consider adding "both" in this sentence to clarify that intermediate and shallow applies to both species (B. napus and H. fulva).
Reply: We added "both" to make this clarification (L648).

.) Line 653: You refer to crops, but use "contrary to our findings". This can be a bit misleading sine you did not study crops. Please rephrase.
Reply: We studied the coffee, which is a perennial crop, planted with shade trees. Therefore, we do not think that the sentence is misleading and requires to be rephrased.

.) Table 4: Two decimals shown for Clay at 5 cm, while all other have one decimal points.
Reply: The correction has been made.

.) You refer to the findings of Poca et al. (2019) when discussing the potentially fractionated coffee xylem water in Figure 3b. However, the mycorrhiza seem to cause a depletion in d2H according to Poca et al. (2019, see their Fig. 3). So, can you really support your findings by Poca's findings?
Reply: We referred to the work of Poca et al. (2019) as an example of the potential influence of mycorrhizal fungi on xylem isotope fractionation. However, we agree that their findings showed a depletion effect rather than an enrichment effect in the deuterium isotopic composition. Therefore, we have decided to remove this citation from the text (L549-550).

.) Figure 8: I recognize that you changed the y-axis to "Proportion of near surface soil water", but the caption is still with "deep soil water".
Reply: The correction has been made.

.) Figure 8: Reconsider if regression lines should be shown. I suggest to leave them out, since the relationship shown is not significant.
Reply: Despite the correlation was not statistically significant, we would like to keep the regression lines in the figure because it helps to highlight the differences across the investigated species.

[revised manuscript text omitted]